# Learning low-dimensional state embeddings and metastable clusters from time series data

**Yifan Sun**
Carnegie Mellon University
yifans@andrew.cmu.edu

**Yaqi Duan**
Princeton University
yaqid@princeton.edu

**Hao Gong**
Princeton University
hgong@princeton.edu

**Mengdi Wang**
Princeton University
mengdiw@princeton.edu

## Abstract

This paper studies how to find compact state embeddings from high-dimensional Markov state trajectories, where the transition kernel has a small intrinsic rank. In the spirit of diffusion map, we propose an efficient method for learning a low-dimensional state embedding and capturing the process's dynamics. This idea also leads to a kernel reshaping method for more accurate nonparametric estimation of the transition function. State embedding can be used to cluster states into metastable sets, thereby identifying the slow dynamics. Sharp statistical error bounds and misclassification rate are proved. Experiment on a simulated dynamical system shows that the state clustering method indeed reveals metastable structures. We also experiment with time series generated by layers of a Deep-Q-Network when playing an Atari game. The embedding method identifies game states to be similar if they share similar future events, even though their raw data are far different.

## 1 Introduction

High-dimensional time series is ubiquitous in scientific studies and machine learning. Finding compact representation from state-transition trajectories is often a prerequisite for uncovering the underlying physics and making accurate predictions. Suppose that we are given a Markov process $\{X_t\}$ taking values in $\Omega \subset \mathbb{R}^d$. Let $p(y|x)$ be the one-step transition density function (transition kernel) of the Markov process. In practice, state-transition trajectories may appear high-dimensional, but they are often generated by a system with fewer internal parameters and small intrinsic dimension.

In this paper, we focus on problems where the transition kernel $p(y|x)$ admits a low-rank decomposition structure. Low-rank or nearly low-rank nature of the transition kernel has been widely identified in scientific and engineering applications, e.g. molecular dynamics [RZMC11, SS13], periodized diffusion process [Går54], traffic transition data [ZW18, DKW19], Markov decision process and reinforcement learning [KAL16]. For reversible dynamical systems, leading eigenfunctions of $p$ are related to metastable sets and slow dynamics [SS13]. Low-rank latent structures also helps state representation learning and dimension reduction in robotics and control [BSB$^+$15].

Our goal is to estimate the transition kernel $p(\cdot|\cdot)$ from finite time series and find state representation in lower dimensions. For nonparametric estimation of probability distributions, one natural approach is the kernel mean embedding (KME). Our approach starts with a kernel space, but we "open up" the kernel function into a set of features. We will leverage the low-rankness of $p$ in the spirit of diffusion map for dimension reduction. By using samples of transition pairs $\{(X_t, X_{t+1})\}$, we can estimate the "projection" of $p$ onto the product feature space and finds its leading singular functions. This allows

us to learn state embeddings that preserve information about the transition dynamics. Our approach can be thought of as a generalization of diffusion map to nonreversible processes and Hilbert space. We show that, when the features can fully express the true $p$, the estimated state embeddings preserve the diffusion distances and can be further used to cluster states that share similar future paths, thereby finding metastable sets and long-term dynamics of the process.

The contributions of this paper are:

1. *KME Reshaping for more accurate estimation of $p$.* The method of KME reshaping is proposed to estimate $p$ from dependent time series data. The method takes advantage of the low-rank structure of $p$ and can be implemented efficiently in compact space. Theorem 1 gives a finite-sample error bound and shows that KME reshaping achieves significantly smaller error than plain KME.

2. *State embedding learning with statistical distortion guarantee.* In light of the diffusion map, we study state embedding by estimating the leading spectrum of the transition kernel. Theorems 2,3 show that the state embedding largely preserves the diffusion distance.

3. *State clustering with misclassification error bound.* Based on the state embeddings, we can further aggregate states to preserve the transition dynamics and find metastable sets. Theorem 4 establishes the statistical misclassification guarantee for continuous-state Markov processes.

4. *Experiments with diffusion process and Atari game.* The first experiment studies a simulated stochastic diffusion process, where the results validate the theoretical bounds and reveals metastable structures of the process. The second experiment studies the time series generated by a deep Q-network (DQN) trained on an Atari game. The raw time series is read from the last hidden layer as the DQN is run. The state embedding results demonstrate distinctive and interpretable clusters of game states. Remarkably, we observe that game states that are close in the embedding space share similar future moves, even if their raw data are far different.

To our best knowledge, our theoretical results on estimating $p$ and state embedding are the first of their kind for continuous-state nonreversible Markov time series. Our methods and analyses leverage spectral properties of the transition kernel. We also provide the first statistical guarantee for partitioning the continuous state space according to diffusion distance.

**Related Work**    Spectral dimension reduction methods find wide use in data analysis and scientific computing. Diffusion map is a prominent dimension reduction tool which applies to data analysis, graph partitioning and dynamical systems [LL06],[CKL$^+$08]. For molecular dynamics, [SNL$^+$11] showed that leading spectrum of transition operator contains information on slow dynamics of the system, and it can be used to identify coresets upon which a coarse-grained Markov state model could be built. [KSM18] extended the transfer operator theory to reproducing kernel spaces and pointed out the these operators are related to conditional mean embeddings of the transition distributions. See [KKS16, KNK$^+$18] for surveys on data-driven dimension reduction methods for dynamical systems. They did not study statistical properties of these methods which motivated our research.

Nonparametric estimation of the Markov transition operator has been thoroughly studied, see [Yak79, Lac07, Sar14]. Among nonparametric methods, kernel mean embeddings are prominent for representing probability distributions [BTA04, SGSS07]. [SHSF09] extended kernel embedding methods to conditional distributions. [GLB$^+$12] proposed to use conditional mean embedding to model Markov decision processes. See [MFSS17] for a survey on kernel mean embedding. None of these works considered low-rank estimation of Markov transition kernel, to our best knowledge.

Estimation of low-rank transition kernel was first considered by [ZW18] in the special case of finite-state Markov chains. [ZW18] used a singular thresholding method to estimate the transition matrix and proves near-optimal error upper and lower bounds. They also proved misclassification rate for state clustering when the chain is lumpable or aggregable. [LWZ18] studied a rank-constrained maximum likelihood estimator of the transition matrix. [DKW19] proposed a novel approach for finding state aggregation by spectral decomposing transition matrix and transforming singular vectors. For continuous-state reversible Markov chains, [LP18] studied the nonparametric estimation of transition kernel via Galerkin projection with spectral thresholding. They proved recovery error bounds when eigenvalues decay exponentially.

**Notations**    For a function $f : \Omega \to \mathbb{R}$, we define $\|f\|_{L^2}^2 := \int_\Omega f(x)^2 dx$ and $\|f\|_{L^2(\pi)}^2 := \int_\Omega \pi(x)f(x)^2 dx$, respectively. For $g(\cdot, \cdot) \to \mathbb{R}$, we define $\|g(\cdot, \cdot)\|_{L^2(\pi)\times L^2} := (\int \pi(x)g(x,y)^2 dydx)^{1/2}$. We use $\|\cdot\|$ to

denote the Euclidean norm of a vector. We let $t_{mix}$ denote the mixing time of the Markov process [LP17], i.e, $t_{mix} = \min\left\{t \mid \text{TV}(P^t(\cdot \mid x), \pi(\cdot)) \leq \frac{1}{4}, \forall x \in \Omega\right\}$, where $TV$ is the total variation divergence between two distributions. Let $\pi(x)$ be the density function of invariant measure of the Markov chain. Let $p(x, y)$ be the density of the invariant measure of the bivariate chain $\{(X_t, X_{t+1})\}_{t=0}^{\infty}$, i.e. $p(X_t, X_{t+1}) = \pi(X_t)p(X_{t+1}|X_t)$. We use $\mathbb{P}(\cdot)$ to denote probability of an event.

## 2  KME Reshaping for Estimating $p$

In this section we study the estimation of of transition function $p$ from a finite trajectory $\{X_t\}_{t=1}^n \subset \mathbb{R}^d$. We make following low-rank assumption regarding the transition kernel $p$, which is key to more accurate estimation.

**Assumption 1.** *There exist real-valued functions $\{u_k\}_{k=1}^r$, $\{v_k\}_{k=1}^r$ on $\Omega$ such that $p(y|x) := \sum_{k=1}^r \sigma_k u_k(x) v_k(y)$, where $r$ is the rank.*

Due to the asymmetry of $p(\cdot, \cdot)$ and lack of reversibility, we use two reproducing kernel Hilbert spaces $\mathcal{H}$ and $\tilde{\mathcal{H}}$ to embed the left and right side of $p$. Let $K$ and $\tilde{K}$ be the kernel functions for $\mathcal{H}$ and $\tilde{\mathcal{H}}$ respectively. The Kernel Mean Embedding (KME) $\mu_p(x, y)$ of the joint distribution $p(x, y)$ into the product space $\mathcal{H} \times \tilde{\mathcal{H}}$ is defined by

$$\mu_p(x, y) := \int K(x, u)\tilde{K}(y, v)p(u, v)dudv.$$

Given sample transition pairs $\{(X_i, X_i')\}_{i=1}^n$, the natural empirical KME estimator is $\tilde{\mu}_p(x, y) = \frac{1}{n}\sum_{i=1}^n K(X_i, x)\tilde{K}(X_i', y)$. If data pairs are independent, one can show that the embedding error $\|\mu_p - \tilde{\mu}_p\|_{\mathcal{H} \times \tilde{\mathcal{H}}}$ is approximately $\sqrt{\frac{\mathbb{E}_{(X,Y)\sim p}[K(X,X)\tilde{K}(Y,Y)]}{n}}$ (Lemma 1 in Appendix). Next we propose a sharper KME estimator.

Suppose that the kernel functions $K$ and $\tilde{K}$ are continuous and symmetric semi-definite. Let $\{\Phi_j(x)\}_{j\in J}$ and $\{\tilde{\Phi}_j(x)\}_{j\in J}$ be the real-valued feature functions on $\Omega$ such that $K(x, y) = \sum_{j\in J}\Phi_j(x)\Phi_j(y)$, and $\tilde{K}(x, y) = \sum_{j\in J}\tilde{\Phi}_j(x)\tilde{\Phi}_j(y)$. In practice, if one is given a shift-invariant symmetric kernel function, we can generate finitely many random Fourier features to approximate the kernel [RR08]. In what follows we assume without loss of generality that $J$ is finite of size $N$.

Let $\Phi(x) = [\Phi_1(x), \ldots, \Phi_N(x)]^T \in \mathbb{R}^N$. We define the "projection" of $p$ onto the feature space by

$$\mathbf{P} = \int p(x, y)\Phi(x)\tilde{\Phi}(y)^T dxdy. \tag{1}$$

Assumption 1 suggests that $\text{rank}(\mathbf{P}) \leq r$ (Lemma 2 in Appendix). Note that the KME of $p(x, y)$ is equivalent to $\mu_p(x, y) = \Phi(x)^T \mathbf{P}\tilde{\Phi}(y)$ (Lemma 3 in Appendix). The matrix $\mathbf{P}$ is of finite dimensions, therefore we can estimate it tractably from the trajectory $\{X_t\}$ by

$$\hat{\mathbf{P}} := \frac{1}{n}\sum_{t=1}^n \Phi(X_t)\tilde{\Phi}(X_{t+1})^T. \tag{2}$$

Since the unknown $\mathbf{P}$ is low-rank, we propose to apply singular value truncation to $\hat{\mathbf{P}}$ for obtaining a better KME estimator. The algorithm is given below:

---
**Algorithm 1:** Reshaping the Kernel Mean Embedding.

---
**Input:**$\{X_1, \ldots, X_n\}, r$;
Get $\hat{\mathbf{P}}$ by (2), compute its SVD: $\hat{\mathbf{P}} = \hat{\mathbf{U}}\hat{\mathbf{\Sigma}}\hat{\mathbf{V}}$;
Let $\tilde{\mathbf{P}} := \hat{\mathbf{U}}\hat{\mathbf{\Sigma}}_{[1\ldots r]}\hat{\mathbf{V}}$ be the best rank $r$ approximation of $\hat{\mathbf{P}}$;
Let $\hat{\mu}_p(x, y) := \Phi(x)^T\tilde{\mathbf{P}}\tilde{\Phi}(y)$;
**Output:** $\hat{\mu}_p(x, y)$

---

We analyze the convergence rate of $\hat{\mu}_p$ to $\mu_p$. Let $K_{max} := \max\{\sup_{x\in\Omega} K(x, x), \sup_{x\in\Omega}\tilde{K}(x, x)\}$. We define following kernel covariance matrices:

$$\mathbf{V}_1 = \mathbb{E}_{(X,Y)\sim p}[\Phi(X)\Phi(X)^T\tilde{K}(Y, Y)], \qquad \mathbf{V}_2 = \mathbb{E}_{(X,Y)\sim p}[K(X, X)\tilde{\Phi}(Y)\tilde{\Phi}(Y)^T].$$

Let $\bar{\lambda} := \max\{\lambda_{\max}(\mathbf{V}_1), \lambda_{\max}(\mathbf{V}_2)\}$. We show the following finite-sample error bound.

**Theorem 1** (KME Reshaping). *Let Assumption 1 hold. For any $\delta \in (0, 1)$, we have*

$$\|\mu_p - \hat{\mu}_p\|_{\mathcal{H} \times \tilde{\mathcal{H}}} = \|\mathbf{P} - \tilde{\mathbf{P}}\|_F \leq C\sqrt{r}\left(\sqrt{\frac{t_{mix}\bar{\lambda}\log(2t_{mix}N/\delta)}{n}} + \frac{t_{mix}K_{\max}\log(2t_{mix}N/\delta)}{3n}\right)$$

*with probability at least $1 - \delta$, where $C$ is a universal constant.*

The KME reshaping method and Theorem 1 enjoys the following advantages:

1. **Improved accuracy compared to plain KME.** The plain KME $\tilde{\mu}_p$'s estimation error is approximately $\sqrt{\frac{\mathbb{E}_{(X,Y)\sim p}[K(X,X)\tilde{K}(Y,Y)]}{n}}$ (Appendix Lemma 1). Note that $\text{Tr}(\mathbf{V}_1) = \text{Tr}(\mathbf{V}_2) = \mathbb{E}_{(X,Y)\sim p}[K(X,X)\tilde{K}(Y,Y)]$. When $r \ll N$, we typically have $r\bar{\lambda} \ll \text{Tr}(\mathbf{V}_1) = \text{Tr}(\mathbf{V}_2)$, therefore the reshaped KME has a significantly smaller estimation error.

2. **Ability to handle dependent data.** Algorithm 1 applies to time series consisting of highly dependent data. The proof of Theorem 1 handles dependency by constructing a special matrix martingale and using the mixing properties of the Markov process to analyze its concentration.

3. **Tractable implementation.** Kernel-based methods usually require memorizing all the data and may be intractable in practice. Our approach is based on a finite number of features and only needs to low-dimensional computation. As pointed out by [RR08], one can approximate any shift-invariant kernel function using $N$ features where $N$ is linear with respect to the input dimension $d$. Therefore Algorithm 1 can be approximately implemented in $O(nd^2)$ time and $O(d^2)$ space.

## 3 Embedding States into Euclidean Space

In this section we want to learn low-dimensional representations of the state space $\Omega$ to capture the transition dynamics. We need following extra assumption that $p$ can be fully represented in the kernel space.

**Assumption 2.** *The transition kernel belongs to the product Hilbert space, i.e., $p(\cdot \mid \cdot) \in \mathcal{H} \times \tilde{\mathcal{H}}$.*

For two arbitrary states $x, y \in \Omega$, we consider their distance given by

$$dist(x, y) := \|p(\cdot|x) - p(\cdot|y)\|_{L^2} = \left(\int \big(p(z|x) - p(z|y)\big)^2 dz\right)^{1/2}. \tag{3}$$

Eq. (3) is known as the diffusion distance [NLCK06]. It measures the similarity between future paths of two states. We are motivated by the diffusion map approach for dimension reduction [LL06, CKL$^+$08, KSM18]. Diffusion map refers to the leading eigenfunctions of the transfer operator of a reversible dynamical system. We will generalize it to nonreversible processes and feature spaces.

For simplicity of presentation, we assume without loss of generality that $\{\Phi_i\}_{i=1}^N$ and $\{\tilde{\Phi}_i\}_{i=1}^N$ are $L_2(\pi)$ and $L_2$ orthogonal bases of $\mathcal{H}$ and $\tilde{\mathcal{H}}$ respectively, with squared norms $\rho_1 \geq \cdots \geq \rho_N$ and $\tilde{\rho}_1 \geq \cdots \geq \tilde{\rho}_N$ respectively. Any given features can be orthogonalized to satisfy this condition. In particular, let the matrix $\mathbf{C} := diag[\rho_1, \cdots, \rho_N]$, $\tilde{\mathbf{C}} := diag[\tilde{\rho}_1, \cdots, \tilde{\rho}_N]$, it is easy to verify that $p(y|x) = \Phi(x)^T\mathbf{C}^{-1}\mathbf{P}\tilde{\mathbf{C}}^{-1}\tilde{\Phi}(y)$. Let $\mathbf{C}^{-1/2}\mathbf{P}\tilde{\mathbf{C}}^{-1/2} = \mathbf{U}^{(\rho)}\mathbf{\Sigma}_{[1\cdots r]}^{(\rho)}\mathbf{V}^{(\rho)}$ be its SVD. We define the state embedding as

$$\mathbf{\Psi}(x) := \left(\Phi(x)^T\mathbf{C}^{-1/2}\mathbf{U}^{(\rho)}\mathbf{\Sigma}_{[1\cdots r]}^{(\rho)}\right)^T.$$

It is straightforward to verify that $dist(x, z) = \|\mathbf{\Psi}(x) - \mathbf{\Psi}(z)\|$. We propose to estimate $\mathbf{\Psi}$ in Algorithm 2.

---

**Algorithm 2:** Learning State Embedding

**Input:** $\{X_1, \ldots, X_n\}, r$;

Get $\hat{\mathbf{P}}$ from (2), compute SVD $\hat{\mathbf{U}}^{(\rho)}\hat{\mathbf{\Sigma}}^{(\rho)}\hat{\mathbf{V}}^{(\rho)} = \mathbf{C}^{-1/2}\hat{\mathbf{P}}\tilde{\mathbf{C}}^{-1/2}$;

Compute state embedding using first $r$ singular pairs $\hat{\mathbf{\Psi}}(x) = \left(\Phi(x)^T\mathbf{C}^{-1/2}\hat{\mathbf{U}}^{(\rho)}\hat{\mathbf{\Sigma}}_{[1\cdots r]}^{(\rho)}\right)^T$;

**Output:** $x \mapsto \hat{\mathbf{\Psi}}(x)$

---

Let $\widehat{dist}(x, z) := \|\hat{\boldsymbol{\Psi}}(x) - \hat{\boldsymbol{\Psi}}(z)\|$. We show that the estimated state embeddings preserve the diffusion distance with an additive distortion.

**Theorem 2** (Maximum additive distortion of state embeddings). *Let Assumptions 1,2 hold. Let $L_{max} := \sup_{x \in \Omega} \Phi(x)^T \boldsymbol{C}^{-1} \Phi(x)$ and let $\kappa$ be the condition number of $\sqrt{\pi(x)} p(y|x)$. For any $0 < \delta < 1$ and for all $x, z \in \Omega$, $|dist(x, z) - \widehat{dist}(x, z)|$ is upper bounded by:*

$$C \sqrt{\frac{L_{\max}}{\rho_N \tilde{\rho}_N}} \left[ \sqrt{2}\kappa + 1 \right] \left( \sqrt{\frac{t_{mix} \bar{\lambda} \log(2 t_{mix} N/\delta)}{n}} + \frac{t_{mix} K_{\max} \log(2 t_{mix} N/\delta)}{3n} \right)$$

*with probability at least $1 - \delta$ for some constant $C$.*

Under Assumption 2, we can recover the full transition kernel from data by

$$\hat{p}(y|x) = \Phi(x)^T \boldsymbol{C}^{-1/2} \hat{\boldsymbol{U}}^{(\rho)} \hat{\boldsymbol{\Sigma}}_{[1\cdots r]}^{(\rho)} (\hat{\boldsymbol{V}}^{(\rho)})^T \tilde{\boldsymbol{C}}^{-1/2} \tilde{\Phi}(y).$$

**Theorem 3** (Recovering the transition density). *Let Assumptions 1,2 hold. For any $\delta \in (0, 1)$,*

$$\|p(\cdot|\cdot) - \hat{p}(\cdot|\cdot)\|_{L^2(\pi) \times L^2} \leq C \sqrt{\frac{r}{\rho_N \tilde{\rho}_N}} \left( \sqrt{\frac{t_{mix} \bar{\lambda} \log(2 t_{mix} N/\delta)}{n}} + \frac{t_{mix} K_{\max} \log(2 t_{mix} N/\delta)}{3n} \right)$$

*with probability at least $1 - \delta$ for some constant $C$.*

Theorems 2,3 provide the first statistical guarantee for learning state embeddings and recovering the transition density for continuous-state low-rank Markov processes. The state embedding learned by Algorithm 2 can be represented in $O(Nr)$ space since $\Phi$ is priorly known. When $\Omega$ is finite and the feature map is identity, Theorem 3 nearly matches the the information-theoretical error lower bound given by [LWZ18].

## 4 Clustering States Using Diffusion Distances

We want to find a partition of the state space into $m$ disjoint sets $\Omega_1 \cdots \Omega_m$. The principle is if $x, y \in \Omega_i$ for some $i$, then $p(\cdot|x) \approx p(\cdot|y)$, meaning that states within the same set share similar future paths. This motivates us to study the following optimization problem, which has been considered in studies for dynamical systems [SS13],

$$\min_{\Omega_1, \cdots, \Omega_m} \min_{q_1 \in \tilde{\mathcal{H}}, \cdots, q_m \in \tilde{\mathcal{H}}} \sum_{i=1}^{m} \int_{\Omega_i} \pi(x) \|p(\cdot|x) - q_i(\cdot)\|_{L^2}^2 dx, \tag{4}$$

We assume without loss of generality that it admits a unique optimal solution, which we denote by $(\Omega_1^*, \ldots, \Omega_m^*)$ and $(q_1^*, \ldots, q_m^*)$. Under Assumption 2, each $q_i^*(\cdot)$ is a probability distribution and can be represented by right singular functions $\{v_k(\cdot)\}_{k=1}^r$ of $p(\cdot|\cdot)$ (Lemma 7 in Appendix). We propose the following state clustering method:

---
**Algorithm 3:** Learning metastable state clusters
---
**Data:** $\{X_1, \ldots, X_n, r, m\}$
Use Alg. 2 to get state embedding $\hat{\boldsymbol{\Psi}} : \Omega \mapsto \mathbb{R}^r$;
Solve k-means problem:

$$\min_{\Omega_1, \cdots, \Omega_m} \min_{s_1 \cdots, s_m \in \mathbb{R}^r} \sum_{i=1}^{m} \int_{\Omega_i} \pi(x) \|\hat{\boldsymbol{\Psi}}(x) - s_i\|^2 dx;$$

**Output:** $\hat{\Omega}_1^* \cdots \hat{\Omega}_m^*$

---

The k-means method uses the invariant measure $\pi$ as a weight function. In practice if $\pi$ is unknown, one can pick any reasonable measure and the theoretical bound can be adapted to that measure.

We analyze the performance of the state clustering method on finite data. Define the misclassification rate as

$$M(\hat{\Omega}_1^*, \cdots, \hat{\Omega}_m^*) := \min_{\sigma} \sum_{j=1}^{m} \frac{\pi(\{x : x \in \Omega_j^*, i \notin \hat{\Omega}_{\sigma(j)}^*\})}{\pi(\Omega_j^*)},$$

where $\sigma$ is taken over all possible permutations over $\{1, \ldots, m\}$. The misclassification rate is always between 0 and $m$. We let $\Delta_1^2 := \min_k \min_{l \neq k} \pi(\Omega_k^*) \|q_l^* - q_k^*\|_{L^2}^2$ and let $\Delta_2^2$ be the minimal value of (4).

**Theorem 4** (Misclassification error bound for state clustering). *Let Assumptions 1,2 hold. Let $\kappa$ be the condition number of $\sqrt{\pi(x)}p(y|x)$. If $\Delta_1 > 4\Delta_2$, then for any $0 < \delta < 1$ and $\epsilon > 0$, by letting*

$$n = \Theta\left( \frac{\kappa^2 r \bar{\lambda} t_{mix} \log(2t_{mix}N/\delta)}{\rho_N \tilde{\rho}_N} \cdot \max\left\{ \frac{1}{(\Delta_1 - 4\Delta_2)^2}, \frac{1}{\epsilon\Delta_1^2}, \frac{\Delta_2^2}{\epsilon^2\Delta_1^4} \right\} \right),$$

*we have $M(\hat{\Omega}_1^*, \cdots, \hat{\Omega}_m^*) \leq \frac{16\Delta_2^2}{\Delta_1^2} + \epsilon$ with probability at least $1 - \delta$.*

The full proof is given in Appendix. The condition $\Delta_1 > 4\Delta_2$ is a separability condition needed for finding the correct clusters with high probability, and $\frac{16\Delta_2^2}{\Delta_1^2}$ is non-vanishing misclassification error. In the case of reversible finite-state Markov process, the clustering problem is equivalent to finding the *optimal metastable m-full partition* given by $\mathrm{argmax}_{\Omega_1, \cdots, \Omega_m} \sum_{k=1}^m p(\Omega_k|\Omega_k)$, where $p(\Omega_j|\Omega_i) := \frac{1}{\pi(\Omega_i)} \int_{x\in\Omega_i, y\in\Omega_j} \pi(x)p(y|x)dydx$ [ELVE08, SS13]. The optimal partition $(\Omega_1^*, \cdots, \Omega_m^*)$ gives metastable sets that can be used to construct a reduced-order Markov state model [SS13]. In the more general case of nonreversible Markov chains, the proposed method will cluster states together if they share similar future paths. It provides an unsupervised learning method for state aggregation, which is a widely used heuristic for dimension reduction of control and reinforcement learning [BT96, SJJ95].

# 5   Experiments

## 5.1   Stochastic Diffusion Processes

We test the proposed approach on simulated diffusion processes of the form $dX_t = -\nabla V(X_t)dt + \sqrt{2}dB_t, X_t \in \mathbb{R}^d$, where $V(\cdot)$ is a potential function and $\{B_t\}_{t\geq 0}$ is the standard Brownian motion. For any interval $\tau > 0$, the discrete-time trajectory $\{X_{k\tau}\}_{k=1}^\infty$ is a Markov process. We apply the Euler method to generate sample path $\{X_{k\tau}\}_{k=1}^n$ according to the stochastic differential equation.

We use the Gaussian kernels $K(x,y) = \tilde{K}(x,y) = \frac{1}{(2\pi\sigma^2)^{d/2}} e^{-\frac{\|x-y\|_2^2}{2\sigma^2}}$ where $\sigma > 0$, and construct RKHS $\mathcal{H} = \tilde{\mathcal{H}}$ from $L^2(\pi)$. To get the features $\Phi$, we generate 2000 random Fourier features $\mathbf{h} = [h_1, h_2, \ldots, h_N]^\top$ such that $K(x,y) \approx \sum_{i=1}^N h_i(x)h_i(y)$ ([RR08]), and then orthogonalize $\mathbf{h}$ to get $\Phi$.

**Comparison between reshaped and plain KME**

We apply Algorithm 1 to find the reshaped KME $\hat{\mu}_p$ and compare its error with the plain KME $\tilde{\mu}_p$ given by (2). The experiment is performed on a four-well diffusion on $\mathbb{R}$, and we take rank $r = 4$. By orthogonalizing $N = 2000$ random Fourier features, we obtain $J = 82$ basis functions. Figure 5.1 shows that the reshaped KME consistently outperforms plain KME with varying sample sizes.

**State clustering to reveal metastable structures**

We apply the state clustering method to analyze metastable structures of a diffusion process whose potential function $V(x)$ is given by Figure 5.1 (a). We generate trajectories of length $n = 10^6$ and take the time interval to be $\tau = 0.1, 1, 5$ and 10. We conduct the state embedding and clustering procedures with rank $r = 4$. Figure 5.1 (c)

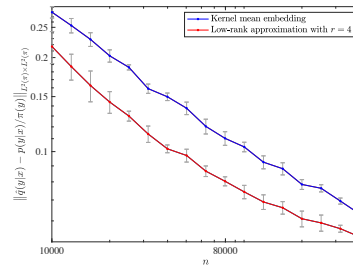

Figure 1: **Reshaped KME versus plain KME.** The error curve approximately satisfies a convergence rate of $n^{-1/2}$.

shows the clustering results for $\tau = 1$ with a varying number of clusters. The partition results reliably reveal metastable sets, which are also known as invariance sets that characterize slow dynamics of this process. Figure 5.1 (d) shows the four-cluster results with varying values of $\tau$, where the contours are based on diffusion distances to the centroid in each cluster. One can see that the diffusion distance contours are dense when $\tau$ takes small values. This is because, when $\tau$ is small, the state embedding method largely captures fast local dynamics. By taking $\tau$ to be larger values, the state embedding

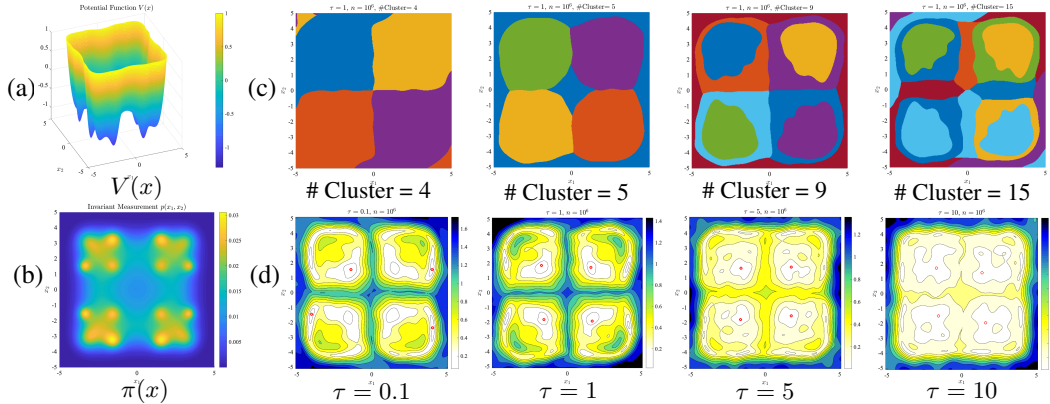

Figure 2: **Metastable state clusters learned from a stochastic diffusion process.** (a) Potential function $V(x)$ of the diffusion process. (b) Invariant measure $\pi(x)$. (c) State clusters based on a state embedding $\hat{\mathbf{\Psi}} : x \mapsto \mathbb{R}^4$. (d) Diffusion distance to the nearest cluster centroid (red dot) illustrated as contour plots.

method begins to capture slower dynamics, which corresponds to low-frequency transitions among the leading metastable sets.

## 5.2   DQN for Demon Attack

We test the state embedding method on the game trajectories of Demon Attack, an Atari 2600 game. In this game, demons appear in waves, move randomly and attack from above, where the player moves to dodge the bullets and shoots with a laser cannon. We train a Deep Q-Network using the architecture given by [MKS+15]. The DQN takes recent image frames of the game as input, processes them through three convolutional layers and two fully connected layers, and outputs a single value for each action, among which the action with the maximal value is chosen. Please refer to Appendix for more details on DQN training. In our experiment, we take the times series generated by the last hidden layer of a trained DQN when playing the game as our raw data. The raw data is a time series of length 47936 and dimension 512, comprising 130 game trajectories. We apply the state embedding method by approximating the Gaussian kernel with 200 random Fourier features. Then we obtain low-dimensional embeddings of the game states in $\mathbb{R}^3$.

**Before embedding vs. after embedding**

Figure 3 visualizes the raw states and the state embeddings using t-SNE, a visualization tool to illustrate multi-dimensional data in two dimensions [VdM08]. In both plots, states that are mapped to nearby points tend to have similar "values" (expected future returns) as predicted by the DQN, as illustrated by colors of data points.

Comparing Figure 3(a) and (b), the raw state data are more scattered, while after embedding they exhibit clearer structures and fewer outliers. The markers $\bigcirc$, $\triangle$, $\diamond$ identify the same pair of game states before and after embedding. They suggest that the embedding method maps game states that are far apart from each other in their raw data to closer neighbors. It can be viewed as a form of compression. The experiment has been repeated multiple times. We consistently observe that state embedding leads to improved clusters and higher granularity in the t-SNE visualization.

**Understanding state embedding from examples**

Figure 4 illustrates three examples that were marked by $\bigcirc$, $\triangle$, $\diamond$ in Figure 3. In each example, we have a pair of game states that were far apart in their raw data but are close to each other after embedding. Also note the two images are visually not alike, therefore any representation learning method based on individual images alone will not consider them to be similar. Let us analyze these three examples:

$\bigcirc$: Both streaming lasers (purple) are about to destroy a demon and generate a reward; Both cannons are moving towards the left end.
$\triangle$: In both images, two new demons are emerging on top of the cannon to join the battle and there is an even closer enemy, leading to future dangers and potential rewards.
$\diamond$: Both cannons are waiting for more targets to appear, and they are both moving towards the center from opposite sides.

These examples suggest that state embedding is able to identify states as similar *if they share similar near-future events and values*, even though they are visually dissimilar and distant from each other in their raw data.

(a) Before Embedding&emsp;&emsp;&emsp;&emsp;&emsp;&emsp;(b) After Embedding

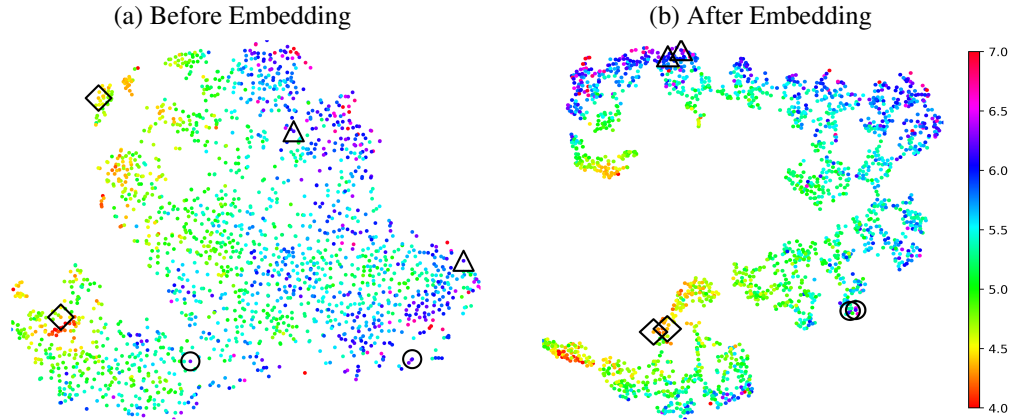

Figure 3: **Visualization of game states before and after embedding in t-SNE plots.** The raw data is a time series of 512 dimensions, which generated by the last hidden layer by the DQN while it is playing Demon Attack. State embeddings are computed from the raw time series using a Gaussian kernel with 200 random Fourier features. Game states are colored by the "value" of the state as predicted by the DQN. The markers $\bigcirc$, $\triangle$, $\diamond$ identify the same pair of game states before and after embedding. Comparing (a) and (b), state embedding improves the granularity of clusters and reveals more structures of the data.

$\bigcirc: V = 6.27$&emsp;$\bigcirc: V = 6.14$&emsp;$\triangle: V = 6.17$&emsp;$\triangle: V = 6.16$&emsp;$\diamond: V = 4.44$&emsp;$\diamond: V = 4.35$

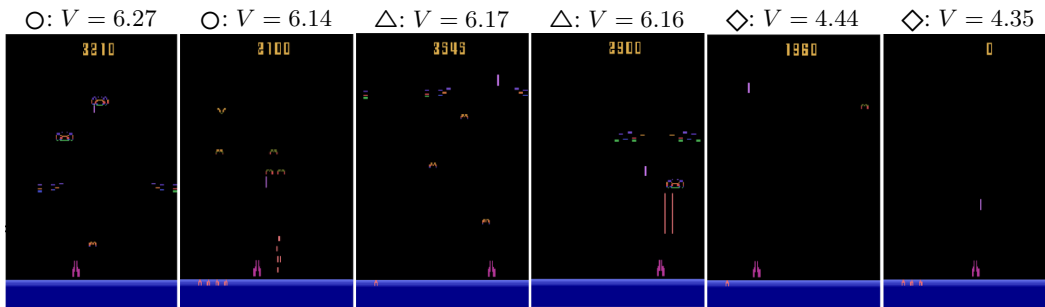

Figure 4: **Pairs of game states that are close after embedding ($\bigcirc$, $\triangle$, $\diamond$ in Figure 3).** Within each pair, the two states share similar "V" values as predicted by the DQN, but they were not close in the raw data and are visually dissimilar. $\bigcirc$: Both streaming lasers (purple) are about to destroy a demon and generate a reward; Both cannons are moving towards the left end. $\triangle$: In both images, two new demons are emerging on top of the cannon to join the battle and there is an even closer enemy, leading to future dangers and potential rewards $\diamond$: Both cannons are waiting for more targets to appear, and they are both moving towards the center from opposite sides. The examples above suggest that state embedding is able to identify states as similar if they share similar near-future paths and values.

**Summary and Future Work**

The experiments validate our theory and lead to interesting discoveries: estimated state embedding captures what would happen in the future conditioned on the current state. Thus the state embedding can be useful to decision makers in terms of gaining insights into the underlying logic of the game, thereby helping them to make better predictions and decisions.

Our methods are inspired by dimension reduction methods from scientific computing and they further leverage the low-rankness of the transition kernel to reduce estimation error and find compact state embeddings. Our theorems provide the basic statistical theory on state embedding/clustering from finite-length dependent time series. They are the first theoretical results known for continuous-state Markov process. We hope our results would motivate more work on this topic and lead to broader applications in scientific data analysis and machine learning. A natural question to ask next is how can one use state embedding to make control and reinforcement learning more efficient. This is a direction for future research.

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
