[Supplementary Material]

# A   Proofs for Section 3

## A .1   Technical Lemmas

**Lemma 1** (Theorem 1 of [LPMST15]). *Suppose* $\{(X_i, X_i')\}_{i=1}^n \overset{i.i.d}{\sim} p(\cdot, \cdot)$, *assume that* $\|f \otimes g\|_\infty \le 1$ *for all* $f \otimes g \in \mathcal{H} \times \tilde{\mathcal{H}}$ *with* $\|f \otimes g\|_{\mathcal{H} \times \tilde{\mathcal{H}}} \le 1$. *Then with probability at least* $1 - \delta$:

$$\|\mu_p - \hat{\mu}_p\|_{\mathcal{H} \times \tilde{\mathcal{H}}} \le 2\sqrt{\frac{\mathbb{E}_{(X,Y) \sim \mathbb{P}}[K(X,X)\tilde{K}(Y,Y)]}{n}} + \sqrt{\frac{2\log(1/\delta)}{n}}.$$

*Proof.* See [LPMST15] for detailed proof. $\square$

**Lemma 2.** *Under Assumption 1, the projection matrix*

$$\mathbf{P} = \int \pi(x) p(y|x) \Phi(x) \tilde{\Phi}(y)^T dxdy$$

*has rank at most* $r$.

*Proof.* We define $N \times 1$ vectors $\vec{u}_k$ and $\vec{v}_k$ as:

$$\vec{u}_k := \int \pi(x) u_k(x) \Phi(x) dx, \tag{1}$$

$$\vec{v}_k := \int v_k(y) \tilde{\Phi}(y) dy. \tag{2}$$

Then under Assumption 1, we can write:

$$\mathbf{P} = \int \pi(x) p(y|x) \Phi(x) \tilde{\Phi}(y)^T dxdy \tag{3}$$

$$= \sum_{k=1}^r \int \sigma_k \pi(x) u_k(x) v_k(y) \Phi(x) \tilde{\Phi}(y) dy \tag{4}$$

$$= \sum_{k=1}^r \sigma_k \vec{u}_k \vec{v}_k^T. \tag{5}$$

$\square$

**Lemma 3.** *Suppose that* $K(x,u) = \Phi(x)^T \Phi(u)$, $\tilde{K}(y,v) = \tilde{\Phi}(v)^T \tilde{\Phi}(y)$. *With* $\mathbf{P}$ *given by*

$$\mathbf{P} = \int p(x,y) \Phi(x) \tilde{\Phi}(y)^T dxdy.$$

*We can represent KME of joint distrubution* $\mu_p(u,v)$ *as:*

$$\mu_p(x,y) = \int K(x,u) \tilde{K}(y,v) p(u,v) dudv = \Phi(x)^T \mathbf{P} \tilde{\Phi}(y).$$

*Proof.* By definition of KME for $p(u,v)$ into $\mathcal{H} \times \tilde{\mathcal{H}}$, we have:

$$\mu_p(x,y) = \int K(x,u) \tilde{K}(y,v) p(u,v) dudv.$$

We then write kernels in terms of features and get $K(x,u) = \Phi(x)^T \Phi(u)$, $\tilde{K}(y,v) = \tilde{\Phi}(v)^T \tilde{\Phi}(y)$. Plugging this into the definition of KME we get:

$$\mu_p(x,y) = \int K(x,u) \tilde{K}(y,v) p(u,v) dudv \tag{6}$$

$$= \int \Phi(x)^T \Phi(u) \tilde{\Phi}(v)^T \Phi(y) p(u,v) dudv \tag{7}$$

$$= \Phi(x)^T \left( \int p(u,v) \Phi(u) \tilde{\Phi}(v)^T dudv \right) \Phi(y) \tag{8}$$

$$= \Phi(x)^T \mathbf{P} \Phi(y). \tag{9}$$

$\square$

**Lemma 4.** *Consider the KME of $p(\cdot|\cdot)$ into $\mathcal{H} \times \tilde{\mathcal{H}}$ where $\mathcal{H}$ has kernel $K(x,y)$ and $\tilde{\mathcal{H}}$ has kernel $\tilde{K}(x,y)$, $K(x,y) = \Phi(x)^T\Phi(y)$ and $\tilde{K}(x,y) = \tilde{\Phi}(x)^T\tilde{\Phi}(y)$. Let $\mathbf{V}_1 = \mathbb{E}_{(X,Y)\sim p}[\Phi(X)\Phi(X)^T\tilde{K}(Y,Y)]$ and $\mathbf{V}_2 = \mathbb{E}_{(X,Y)\sim p}[K(X,X)\tilde{\Phi}(Y)\tilde{\Phi}(Y)^T]$, $\bar{\lambda} = \max\{\lambda_{\max}(\mathbf{V}_1), \lambda_{\max}(\mathbf{V}_2)\}$. Let $\tau(\epsilon) := \min\{t \mid TV(P^t(\cdot|x), \pi(\cdot)) \le \epsilon, \forall x \in \Omega\}$ denote the mixing time. Let $\epsilon > 0$, $\alpha(\epsilon) = \tau(\frac{\epsilon}{2K_{max}} \wedge \frac{\bar{\lambda}}{K_{max}^2}) + 1$, then:*

$$\mathbb{P}\big(\|\hat{\mathbf{P}} - \mathbf{P}\| \ge \epsilon\big) \le 2\alpha(\epsilon)N\exp\left(-\frac{n\epsilon^2/8}{2\alpha(\epsilon)\bar{\lambda} + \epsilon\alpha(\epsilon)K_{\max}/6}\right). \tag{10}$$

*For any $0 < \delta < 1$, we have*

$$\|\mathbf{P} - \hat{\mathbf{P}}\| \le C\left(\sqrt{\frac{t_{mix}\bar{\lambda}\log(2t_{mix}N/\delta)}{n}} + \frac{t_{mix}K_{\max}\log(2t_{mix}N/\delta)}{3n}\right) \tag{11}$$

*with probability at least $1 - \delta$, where $C$ is an universal constant.*

*Proof.* Let $n_0 = \lfloor n/\alpha \rfloor$. Define

$$\hat{\mathbf{P}}_t := \Phi(X_{t-1})\tilde{\Phi}(X_t)^T \qquad \text{for } t = 2, 3, \ldots, n,$$

and the "thin" sequence:

$$\check{\mathbf{P}}_k^{(l)} := \hat{\mathbf{P}}_{k\alpha+l} - \mathbb{E}(\hat{\mathbf{P}}_{k\alpha+l}|\hat{\mathbf{P}}_{(k-1)\alpha+l}). \tag{12}$$

We first bound $\|\hat{\mathbf{P}}_{k\alpha+l}\|$:

$$\|\hat{\mathbf{P}}_{k\alpha+l}\| = \sup_{\substack{\|\mathbf{v}\|_{\ell_2}\le 1 \\ \|\mathbf{u}\|_{\ell_2}\le 1}} \mathbf{v}^T\hat{\mathbf{P}}_{k\alpha+l}\mathbf{u} \tag{13}$$

$$= \sup_{\substack{\|\mathbf{v}\|_{\ell_2}\le 1 \\ \|\mathbf{u}\|_{\ell_2}\le 1}} \sum_{i,j=1}^{N} \mathbf{v}_i\Phi_i(X_{k\alpha+l-1})\tilde{\Phi}_j(X_{k\alpha+l})\mathbf{u}_j \tag{14}$$

$$\le K_{max}. \tag{15}$$

In the last inequality we use the fact that for any vector $\mathbf{v}$ with $\|\mathbf{v}\|_{\ell_2} \le 1$,

$$\sum_{i=1}^{N} \mathbf{v}_i\Phi_i(x) \le \|\mathbf{v}\|_{\ell_2}\|\Phi(x)\|_{\ell_2} \le \sqrt{K(x,x)} \le K_{max}^{1/2}, \tag{16}$$

where we used $\|\Phi(x)\|_{\ell_2}^2 = K(x,x)$ and the definition of $K_{max}$. Similar argument applies for $\sum_{j=1}^{N} \mathbf{u}_j\Phi_j(x)$. Hence $\|\mathbb{E}(\hat{\mathbf{P}}_{k\alpha+l}|\hat{\mathbf{P}}_{(k-1)\alpha+l})\| \le \mathbb{E}(\|\hat{\mathbf{P}}_{k\alpha+l}\|\|\hat{\mathbf{P}}_{(k-1)\alpha+l}) \le K_{max}$ and we have:

$$\|\check{\mathbf{P}}_k^{(l)}\| \le 2K_{max}. \tag{17}$$

Let $p^\alpha(x|x_{(k-1)\alpha+l})$ be the $(\alpha - 1)$-step transition density starting from $x_{(k-1)\alpha+l}$. We use " · " to denote dot product of two vectors, then we have:

$$(\hat{\mathbf{P}}_{k\alpha+l}\hat{\mathbf{P}}_{k\alpha+l}^T)_{ij} := (\hat{\mathbf{P}}_{k\alpha+l})_{[i,:]} \cdot (\hat{\mathbf{P}}_{k\alpha+l})_{[j,:]} \tag{18}$$

$$= \sum_{p=1}^{N} \Phi_i(X_{k\alpha+l-1})\Phi_j(X_{k\alpha+l-1})\tilde{\Phi}_p^2(X_{k\alpha+l}). \tag{19}$$

Taking conditional expectation yields:

$$\mathbb{E}\left(\hat{\mathbf{P}}_{k\alpha+l}\hat{\mathbf{P}}_{k\alpha+l}^T|x_{(k-1)\alpha+l}\right)_{ij} = \int \Phi_i(x)\Phi_j(x)p^\alpha(x|x_{(k-1)\alpha+1})p(y|x)\sum_{p=1}^{N}\tilde{\Phi}_p^2(y)dxdy \tag{20}$$

$$= \int \Phi_i(x)\Phi_j(x)p^\alpha(x|x_{(k-1)\alpha+1})p(y|x)\tilde{K}(y,y)dxdy. \tag{21}$$

We can write

$$\mathbb{E}\left(\hat{\mathbf{P}}_{k\alpha+l}\hat{\mathbf{P}}_{k\alpha+l}^T|x_{(k-1)\alpha+l}\right)_{ij} = \left(\mathbb{E}\left(\hat{\mathbf{P}}_{k\alpha+l}\hat{\mathbf{P}}_{k\alpha+l}^T|x_{(k-1)\alpha+l}\right) - \mathbf{V}_1\right)_{ij} + (\mathbf{V}_1)_{ij} \tag{22}$$

$$= \int \Phi_i(x)\Phi_j(x)\left(p^\alpha(x|x_{(k-1)\alpha+1}) - \pi(x)\right)p(y|x)\tilde{K}(y,y)dxdy \tag{23}$$

$$+ \int \Phi_i(x)\Phi_j(x)\pi(x)p(y|x)\tilde{K}(y,y)dxdy. \tag{24}$$

Therefore:

$$\|\mathbb{E}\left(\hat{\mathbf{P}}_{k\alpha+l}\hat{\mathbf{P}}_{k\alpha+l}^T|x_{(k-1)\alpha+l}\right)\| \tag{25}$$

$$= \|\left(\mathbb{E}\left(\hat{\mathbf{P}}_{k\alpha+l}\hat{\mathbf{P}}_{k\alpha+l}^T|x_{(k-1)\alpha+l}\right) - \mathbf{V}_1\right) + \mathbf{V}_1\| \tag{26}$$

$$\le \|\left(\mathbb{E}\left(\hat{\mathbf{P}}_{k\alpha+l}\hat{\mathbf{P}}_{k\alpha+l}^T|x_{(k-1)\alpha+l}\right) - \mathbf{V}_1\| + \|\mathbf{V}_1\| \tag{27}$$

$$= \underbrace{\sup_{\substack{\|\mathbf{v}\|_{\ell_2}\le 1, \\ \|\mathbf{u}\|_{\ell_2}\le 1}} \int \left[\mathbf{v}^T\Phi(x)\right]\left[\mathbf{u}^T\Phi(x)\right]p^\alpha(x|x_{(k-1)\alpha+l} - \pi(x))p(y|x)\tilde{K}(y,y)dydx}_{T_1} + \|\mathbf{V}_1\|. \tag{28}$$

By (16) we have $\|\mathbf{v}^T\Phi\|_\infty \le K_{max}^{1/2}$ and similarly $\|\mathbf{u}^T\Phi\|_\infty \le K_{max}^{1/2}$, using definition of $K_{max}$ we have $\tilde{K}(y,y) \le K_{max}$, we bound the term $T_1$ by:

$$T_1 = \sup_{\substack{\|\mathbf{v}\|_{\ell_2}\le 1, \\ \|\mathbf{u}\|_{\ell_2}\le 1}} \int \left[\mathbf{v}^T\Phi(x)\right]\left[\mathbf{u}^T\Phi(x)\right]\left(p^\alpha(x|x_{(k-1)\alpha+l}) - \pi(x)\right)p(y|x)\tilde{K}(y,y)dxdy \tag{29}$$

$$\le K_{max}^2 \int |p^\alpha(x|x_{(k-1)\alpha+l}) - \pi(x)|dx \tag{30}$$

$$\le \bar{\lambda}, \tag{31}$$

where in the last inequality we use the definition of $\alpha(\epsilon)$. By definition of $\bar{\lambda}$ we have $\|\mathbf{V}_1\| \le \bar{\lambda}$, therefore by (28) we get:

$$\|\mathbb{E}\left(\hat{\mathbf{P}}_{k\alpha+l}\hat{\mathbf{P}}_{k\alpha+l}^T|x_{(k-1)\alpha+l}\right)\| \le 2\bar{\lambda}. \tag{32}$$

Similarly, we have:

$$\|\mathbb{E}\left(\hat{\mathbf{P}}_{k\alpha+l}^T\hat{\mathbf{P}}_{k\alpha+l}|x_{(k-1)\alpha+l}\right)\| \le 2\bar{\lambda}. \tag{33}$$

Combining the fact that:

$$0 \preceq \mathbb{E}\left(\check{\mathbf{P}}_k^{(l)}(\check{\mathbf{P}}_k^{(l)})^T|\check{\mathbf{P}}_{k-1}^{(l)}\right) \tag{34}$$

$$= \mathbb{E}\left(\hat{\mathbf{P}}_{k\alpha+l}\hat{\mathbf{P}}_{k\alpha+l}^T|x_{(k-1)\alpha+l}\right) - \mathbb{E}\left(\hat{\mathbf{P}}_{k\alpha+l}|x_{(k-1)\alpha+l}\right)\mathbb{E}\left(\hat{\mathbf{P}}_{k\alpha+l}|x_{(k-1)\alpha+l}\right) \tag{35}$$

$$\preceq \mathbb{E}\left(\hat{\mathbf{P}}_{k\alpha+l}\hat{\mathbf{P}}_{k\alpha+l}^T|x_{(k-1)\alpha+l}\right), \tag{36}$$

we have

$$\|\mathbb{E}\left(\check{\mathbf{P}}_k^{(l)}(\check{\mathbf{P}}_k^{(l)})^T|\check{\mathbf{P}}_{k-1}^{(l)}\right)\| \le \|\mathbb{E}\left(\hat{\mathbf{P}}_{k\alpha+l}\hat{\mathbf{P}}_{k\alpha+l}^T|x_{(k-1)\alpha+l}\right)\| \le 2\bar{\lambda}. \tag{37}$$

Similar argument yields:

$$\|\mathbb{E}\left((\check{\mathbf{P}}_k^{(l)})^T \check{\mathbf{P}}_k^{(l)} | \check{\mathbf{P}}_{k-1}^{(l)}\right)\| \leq 2\bar{\lambda}. \tag{38}$$

Therefore for $1 \leq k \leq n_0$, $1 \leq l \leq \alpha$:

$$\max\left\{\|\mathbb{E}\left(\check{\mathbf{P}}_k^{(l)}(\check{\mathbf{P}}_k^{(l)})^T | \check{\mathbf{P}}_{k-1}^{(l)}\right)\|, \|\mathbb{E}\left((\check{\mathbf{P}}_k^{(l)})^T \check{\mathbf{P}}_k^{(l)} | \check{\mathbf{P}}_{k-1}^{(l)}\right)\|\right\} \leq 2\bar{\lambda}. \tag{39}$$

The norm of predictable quadratic variation process of the matrix martingale $\{\check{\mathbf{P}}_k^{(l)}\}_{k=1}^{n_0}$ can be bounded by:

$$\|\sum_{k=1}^{n_0} \mathbb{E}\left(\check{\mathbf{P}}_k^{(l)}(\check{\mathbf{P}}_k^{(l)})^T | \check{\mathbf{P}}_{k-1}^{(l)}\right)\| \leq \sum_{k=1}^{n_0} \|\mathbb{E}\left(\check{\mathbf{P}}_k^{(l)}(\check{\mathbf{P}}_k^{(l)})^T | \check{\mathbf{P}}_{k-1}^{(l)}\right)\| \leq 2n_0\bar{\lambda}, \tag{40}$$

$$\|\sum_{k=1}^{n_0} \mathbb{E}\left((\check{\mathbf{P}}_k^{(l)})^T \check{\mathbf{P}}_k^{(l)} | \check{\mathbf{P}}_{k-1}^{(l)}\right)\| \leq \sum_{k=1}^{n_0} \|\mathbb{E}\left((\check{\mathbf{P}}_k^{(l)})^T \check{\mathbf{P}}_k^{(l)} | \check{\mathbf{P}}_{k-1}^{(l)}\right)\| \leq 2n_0\bar{\lambda}. \tag{41}$$

By Matrix Freedman Inequality [Tro11], we have:

$$\mathbb{P}(\|\frac{1}{n_0}\sum_{k=1}^{n_0} \check{\mathbf{P}}_k^{(l)}\| \geq \epsilon/2) \leq 2N\exp(-\frac{(\epsilon n_0)^2/8}{2n_0\bar{\lambda} + \epsilon K_{\max} n_0/6}). \tag{42}$$

Next we note that:

$$\left(\mathbb{E}(\hat{\mathbf{P}}_{k\alpha+l}|x_{(k-1)\alpha+l}) - \mathbf{P}\right)_{i,j} = \int \Phi_i(x)\tilde{\Phi}_j(y)p(y|x)\left(p^\alpha(x|x_{(k-1)\alpha+l}) - \pi(x)\right)dxdy, \tag{43}$$

$$\|\mathbb{E}(\hat{\mathbf{P}}_{k\alpha+l}|x_{(k-1)\alpha+l}) - \mathbf{P}\|$$

$$= \sup_{\substack{\|\mathbf{v}\|_{\ell_2}\leq 1 \\ \|\mathbf{u}\|_{\ell_2}\leq 1}} \mathbf{v}^T\left(\mathbb{E}(\hat{\mathbf{P}}_{k\alpha+l}|x_{(k-1)\alpha+l}) - \mathbf{P}\right)\mathbf{u} \tag{44}$$

$$= \sup_{\substack{\|\mathbf{v}\|_{\ell_2}\leq 1, \\ \|\mathbf{u}\|_{\ell_2}\leq 1}} \int \left[\mathbf{v}^T\Phi(x)\right]\left[\mathbf{u}^T\tilde{\Phi}(y)\right]p(y|x)\left(p^\alpha(x|x_{(k-1)\alpha+1}) - \pi(x)\right)dxdy \tag{45}$$

$$\leq K_{\max}\int |p^\alpha(x|x_{(k-1)\alpha+1}) - \pi(x)|dx \tag{46}$$

$$\leq \epsilon/2, \tag{47}$$

where we used definition of mixing-time $\alpha(\epsilon)$ to get last inequality. Then combining (42) and (47) we get:

$$\mathbb{P}(\|\frac{1}{n_0}\sum_{k=1}^{n_0} \hat{\mathbf{P}}_{k\alpha+l} - \mathbf{P}\| \geq \epsilon) \leq 2N\exp(-\frac{(\epsilon n_0)^2/8}{2n_0\bar{\lambda} + \epsilon K_{\max} n_0/6}). \tag{48}$$

Using union bound we get:

$$\mathbb{P}(\|\hat{\mathbf{P}} - \mathbf{P}\| \geq \epsilon) = \mathbb{P}\left(\|\frac{1}{\alpha}\sum_{l=1}^\alpha \frac{1}{n_0}\sum_{k=1}^{n_0} \mathbf{P}_{k\alpha+l} - \mathbf{P}\| \geq \epsilon\right) \tag{49}$$

$$\leq \mathbb{P}\left(\max_{1\leq l\leq\alpha}\|\frac{1}{n_0}\sum_{k=1}^{n_0} \mathbf{P}_{k\alpha+l} - \mathbf{P}\| \geq \epsilon\right) \tag{50}$$

$$\leq \alpha(\epsilon)\max_{1\leq l\leq\alpha}\mathbb{P}\left(\|\frac{1}{n_0}\sum_{k=1}^{n_0} \mathbf{P}_{k\alpha+l} - \mathbf{P}\| \geq t\right) \tag{51}$$

$$\leq 2\alpha(\epsilon)N\exp(-\frac{n\epsilon^2/8}{2\bar{\lambda}\alpha(\epsilon) + \epsilon K_{\max}\alpha(\epsilon)/6}). \tag{52}$$

To get Eqn.(11) from (10), we let $u = \log(\delta)$ and write $\epsilon$ using $u$, then we need:

$$\log\left(2\alpha(\epsilon)N\right) - \frac{n\epsilon^2/8}{\alpha(\epsilon)(2\bar{\lambda} + \epsilon K_{\max}/6)} \leq u, \tag{53}$$

using the fact that $\alpha(\epsilon)$ grows logarithmically (Lemma 5 in [ZW18]) and solve the above inequality for $\epsilon$, we get (11). □

**Lemma 5.** *Under Assumption 1, let $V$ and $\tilde{V}$ be the right singular matrix of $\mathbf{P}$ and $\tilde{\mathbf{P}}$ respectively, then we have:*

$$\inf_{\boldsymbol{O} \in \mathbb{O}_{r \times r}} \|\boldsymbol{V O} - \tilde{\boldsymbol{V}}\| \leq \frac{C}{\sigma_r(\mathbf{P})} \left( \sqrt{\frac{t_{mix}\bar{\lambda}\log(2t_{mix}N/\delta)}{n}} + \frac{t_{mix}K_{\max}\log(2t_{mix}N/\delta)}{3n} \right) \tag{54}$$

*with probability at least $1 - \delta$, $\mathbb{O}$ is the set of all $r \times r$ orthogonal matrices.*

*Proof.* By Lemma.1 of [ZC18], we know:

$$\inf_{\mathbf{O} \in \mathbb{O}_{r \times r}} \|\mathbf{VO} - \tilde{\mathbf{V}}\| \leq \sqrt{2}\|sin\Theta(\mathbf{V}, \tilde{\mathbf{V}})\|. \tag{55}$$

Since $\tilde{\mathbf{V}}$ is also the right singular matrix of $\hat{\mathbf{P}}$, using Wedin's lemma [Wed72] we know:

$$\|sin\Theta(\mathbf{V}, \tilde{\mathbf{V}})\| \leq \frac{\|\mathbf{P} - \hat{\mathbf{P}}\|}{\sigma_r(\mathbf{P})}. \tag{56}$$

Combining above two inequalities and Eqn.(11) finishes the proof. □

## A .2  Proof of Theorem 1

*Proof.* We consider the KME $\mu_p(\cdot, \cdot) \in \mathcal{H} \times \tilde{\mathcal{H}}$ where $\mathcal{H} \times \tilde{\mathcal{H}}$ is the product RKHS, $K(x, y) = \Phi(x)^T \Phi(y)$ is the kernel of $\mathcal{H}$ and $\tilde{K}(x, y) = \tilde{\Phi}(x)^T \tilde{\Phi}(y)$ is the kernel of $\tilde{\mathcal{H}}$. Suppose that $\{\Phi_i^\circ\}_{i=1}^M$ and $\{\tilde{\Phi}_i^\circ\}_{i=1}^{\tilde{M}}$ are respectively the orthonormal bases of $\mathcal{H}$ and $\tilde{\mathcal{H}}$. $\{\Phi_i^\circ\}_{i=1}^M$ is an orthonormal basis for $\mathcal{H}$ means

$$\langle \Phi_i^\circ, \Phi_j^\circ \rangle_{\mathcal{H}} = \begin{cases} 1, & i = j, \\ 0, & i \neq j. \end{cases} \tag{57}$$

Then there exist matrices $\mathbf{W} \in \mathbb{R}^{N \times M}$ and $\tilde{\mathbf{W}} \in \mathbb{R}^{N \times \tilde{M}}$ such that

$$\Phi(\cdot) = \mathbf{W}\Phi^\circ(\cdot), \qquad \tilde{\Phi}(\cdot) = \tilde{\mathbf{W}}\tilde{\Phi}^\circ(\cdot). \tag{58}$$

Note that $\{\Phi_i^\circ\}_{i=1}^M$ is an orthonormal basis of $\mathcal{H}$ implies that $K(x, y) = [\Phi^\circ(x)]^T[\Phi^\circ(y)]$, this is because $K(x, \cdot) \in \mathcal{H}$, so we can write it as:

$$K(x, \cdot) = \sum_{i=1}^M a_{x,i}\Phi_i^\circ(\cdot), \tag{59}$$

where $a_{x,i}$ is the coefficient that depends on $x$. By reproducing property and above identity:

$$\Phi_j^\circ(x) = \langle K(x, \cdot), \Phi_j^\circ(\cdot) \rangle_{\mathcal{H}} \tag{60}$$

$$= \sum_{i=1}^M a_{x,i}\langle \Phi_i^\circ, \Phi_j^\circ \rangle_{\mathcal{H}} \tag{61}$$

$$= a_{x,j}. \tag{62}$$

Plug into (59):

$$K(x, \cdot) = \sum_{i=1}^M a_{x,i}\Phi_i^\circ(\cdot) = \sum_{i=1}^M \Phi_i^\circ(x)\Phi_i^\circ(\cdot). \tag{63}$$

Note that $K(x,y) = [\Phi^\circ(x)]^T \Phi^\circ(y)$ and $K(x,y) = [\Phi(x)]^T \Phi(y) = [\Phi^\circ(x)]^T \mathbf{W}^T \mathbf{W} \Phi^\circ(y)$, therefore,

$$[\Phi^\circ(x)]^T \Phi^\circ(y) = [\Phi^\circ(x)]^T \mathbf{W}^T \mathbf{W} \Phi^\circ(y). \tag{64}$$

We can take $x_1, x_2, \ldots, x_M \in \Omega$ such that

$$\boldsymbol{\Phi}^\circ = [\Phi^\circ(x_1), \Phi^\circ(x_2), \cdots, \Phi^\circ(x_1)]$$

is a non-singular matrix. Then (64) implies

$$(\boldsymbol{\Phi}^\circ)^T \boldsymbol{\Phi}^\circ = (\boldsymbol{\Phi}^\circ)^T \mathbf{W}^T \mathbf{W} \boldsymbol{\Phi}^\circ,$$

therefore,

$$\mathbf{W}^T \mathbf{W} = \mathbf{I}_M. \tag{65}$$

Similar arguments imply

$$\tilde{\mathbf{W}}^T \tilde{\mathbf{W}} = \mathbf{I}_{\tilde{M}}. \tag{66}$$

If a matrix $\mathbf{M} \in \mathbb{R}^{N \times N}$ satisfies $\mathbf{M} = \mathbf{W} \mathbf{M}^\circ \tilde{\mathbf{W}}^T$ for some $\mathbf{M}^\circ \in \mathbb{R}^{M \times \tilde{M}}$, then

$$\|\mathbf{M}\|_F^2 = Tr(\mathbf{M}^T \mathbf{M}) = Tr\left((\mathbf{W} \mathbf{M}^\circ \tilde{\mathbf{W}}^T)^T (\mathbf{W} \mathbf{M}^\circ \tilde{\mathbf{W}}^T)\right) \tag{67}$$

$$= Tr\left((\mathbf{M}^\circ)^T (\mathbf{W}^T \mathbf{W}) \mathbf{M}^\circ (\tilde{\mathbf{W}}^T \tilde{\mathbf{W}})\right) \tag{68}$$

$$= Tr\left((\mathbf{M}^\circ)^T \mathbf{M}^\circ\right) \tag{69}$$

$$= \|\mathbf{M}^\circ\|_F^2. \tag{70}$$

Recall the inner product on $\mathcal{H} \times \tilde{\mathcal{H}}$ is given by the inner product on $\mathcal{H}$ and $\tilde{\mathcal{H}}$: for any $f_1 \otimes g_1 \in \mathcal{H} \times \tilde{\mathcal{H}}$ and $f_2 \otimes g_2 \in \mathcal{H} \times \tilde{\mathcal{H}}$,

$$\langle f_1 \otimes g_1, f_2 \otimes g_2 \rangle_{\mathcal{H} \times \tilde{\mathcal{H}}} = \langle f_1, f_2 \rangle_{\mathcal{H}} \langle g_1, g_2 \rangle_{\tilde{\mathcal{H}}}.$$

It follows that

$$\left\|[\Phi(\cdot)]^T \mathbf{M} \tilde{\Phi}(\cdot)\right\|_{\mathcal{H} \times \tilde{\mathcal{H}}}^2 = \left\|[\Phi^\circ(\cdot)]^T \mathbf{W}^T \mathbf{M} \tilde{\mathbf{W}} \tilde{\Phi}^\circ(\cdot)\right\|_{\mathcal{H} \times \tilde{\mathcal{H}}}^2 = \left\|[\Phi^\circ(\cdot)]^T \mathbf{M}^\circ \tilde{\Phi}^\circ(\cdot)\right\|_{\mathcal{H} \times \tilde{\mathcal{H}}}^2 \tag{71}$$

$$= \left\langle \sum_{i=1}^M \sum_{j=1}^{\tilde{M}} \Phi_i^\circ(\cdot) \mathbf{M}_{ij}^\circ \tilde{\Phi}_j(\cdot), \sum_{k=1}^M \sum_{p=1}^{\tilde{M}} \Phi_k^\circ(\cdot) \mathbf{M}_{kp}^\circ \tilde{\Phi}_p(\cdot) \right\rangle_{\mathcal{H} \times \tilde{\mathcal{H}}} \tag{72}$$

$$= \sum_{i,k=1}^M \sum_{j,p=1}^{\tilde{M}} \mathbf{M}_{ij}^\circ \mathbf{M}_{kp}^\circ \langle \Phi_i^\circ, \Phi_k^\circ \rangle_{\mathcal{H}} \langle \tilde{\Phi}_j^\circ, \tilde{\Phi}_p^\circ \rangle_{\tilde{\mathcal{H}}} \tag{73}$$

$$= \sum_{i=1}^M \sum_{j=1}^{\tilde{M}} (\mathbf{M}_{ij}^\circ)^2 \tag{74}$$

$$= \|\mathbf{M}^\circ\|_F^2 = \|\mathbf{M}\|_F^2. \tag{75}$$

We can conclude that

$$\left\|\Phi(\cdot)^T \mathbf{M} \tilde{\Phi}(\cdot)\right\|_{\mathcal{H} \times \tilde{\mathcal{H}}} = \|\mathbf{M}\|_F. \tag{76}$$

The matrix $\mathbf{P}$ in our paper satisfies

$$\mathbf{P} = \int_{\Omega \times \Omega} p(x,y) \Phi(x) [\tilde{\Phi}(y)]^T dx dy = \mathbf{W} \left( \int_{\Omega \times \Omega} p(x,y) \Phi^\circ(x) [\tilde{\Phi}^\circ(y)]^T dx dy \right) \tilde{\mathbf{W}}^T. \tag{77}$$

We also have

$$\hat{\mathbf{P}} = \frac{1}{n} \sum_{t=1}^n \Phi(X_t) [\tilde{\Phi}(X_{t+1})]^T = \mathbf{W} \left( \frac{1}{n} \sum_{t=1}^n \Phi^\circ(X_t) [\tilde{\Phi}^\circ(X_{t+1})]^T \right) \tilde{\mathbf{W}}^T. \tag{78}$$

Therefore, $\hat{\mathbf{U}} = \mathbf{W}\boldsymbol{\Gamma}$ for some $\boldsymbol{\Gamma} \in \mathbb{R}^{M \times N}$ and $\hat{\mathbf{V}} = \tilde{\mathbf{W}}\tilde{\boldsymbol{\Gamma}}$ for some $\tilde{\boldsymbol{\Gamma}} \in \mathbb{R}^{\tilde{M} \times N}$. It further implies

$$\tilde{\mathbf{P}} = \hat{\mathbf{U}}\hat{\boldsymbol{\Sigma}}_{[1...r]}\hat{\mathbf{V}}^T = \mathbf{W}\big(\boldsymbol{\Gamma}\hat{\boldsymbol{\Sigma}}_{[1...r]}\tilde{\boldsymbol{\Gamma}}^T\big)\tilde{\mathbf{W}}^T. \tag{79}$$

Then using (76) we know that:

$$\|\mu_p - \hat{\mu}_p\|_{\mathcal{H} \times \tilde{\mathcal{H}}} = \|\mathbf{P} - \tilde{\mathbf{P}}\|_F \le \sqrt{2r}\|\mathbf{P} - \tilde{\mathbf{P}}\|, \tag{80}$$

where the inequality follows the fact that $\mathbf{P}$ and $\tilde{\mathbf{P}}$ are both of rank at most $r$ hence $\mathbf{P} - \tilde{\mathbf{P}}$ has rank at most $2r$. According to Weyl's inequality [Wey12], $\|\hat{\mathbf{P}} - \tilde{\mathbf{P}}\| = \sigma_{r+1}(\hat{\mathbf{P}}) \le \|\hat{\mathbf{P}} - \mathbf{P}\|$. It follows that

$$\|\mathbf{P} - \tilde{\mathbf{P}}\| \le \|\mathbf{P} - \hat{\mathbf{P}}\| + \|\hat{\mathbf{P}} - \tilde{\mathbf{P}}\| \le 2\|\hat{\mathbf{P}} - \mathbf{P}\|.$$

Using Eqn.(11) we finish the proof. $\qquad\square$

## B  Proof of Results in Section 4

### B.1  Representation of $p(\cdot|\cdot)$

**Lemma 6.** *Under Assumption 1-2, $p(\cdot|\cdot)$ has following representation:*

$$p(y|x) = \Phi(x)^T \mathbf{C}^{-1}\mathbf{P}\tilde{\mathbf{C}}^{-1}\tilde{\Phi}(y). \tag{81}$$

*where $\mathbf{P} := \int \pi(x)p(y|x)\Phi(x)\tilde{\Phi}(y)^T dxdy$, $\mathbf{C} := diag[\rho_1, \cdots, \rho_N]$ and $\tilde{\mathbf{C}} := diag[\tilde{\rho}_1, \cdots, \tilde{\rho}_N]$.*

*Proof.* We know that $\Upsilon(\cdot) := \mathbf{C}^{-1/2}\Phi(\cdot)$ is a vector of orthonormal functions in $L^2(\pi)$, and $\tilde{\Upsilon}(\cdot) := \tilde{\mathbf{C}}^{-1/2}\tilde{\Phi}(\cdot)$ is a vector of orthonormal functions in $L^2$. Then the coefficient matrix of $p(\cdot|\cdot)$ in expansion under $L^2(\pi) \times L^2$ inner product is given by:

$$\int \pi(x)p(y|x)\mathbf{C}^{-1/2}\Phi(x)\tilde{\Phi}(y)^T\tilde{\mathbf{C}}^{-1/2}dxdy = \mathbf{C}^{-1/2}\bigg(\int \pi(x)p(y|x)\Phi(x)\tilde{\Phi}(y)^T dxdy\bigg)\tilde{\mathbf{C}}^{-1/2} \tag{82}$$

$$= \mathbf{C}^{-1/2}\mathbf{P}\tilde{\mathbf{C}}^{-1/2}. \tag{83}$$

Then we have:

$$p(y|x) = \Upsilon(x)^T \mathbf{C}^{-1/2}\mathbf{P}\tilde{\mathbf{C}}^{-1/2}\tilde{\Upsilon}(y) \tag{84}$$

$$= \Phi(x)^T \mathbf{C}^{-1}\mathbf{P}\tilde{\mathbf{C}}^{-1}\Phi(y). \tag{85}$$

$\qquad\square$

### B.2  Proof of Theorem 2

*Proof.* To simplify notation, denote $\mathbf{R} := \mathbf{C}^{-1/2}\mathbf{P}\tilde{\mathbf{C}}^{-1/2}$. Let $\mathbf{U}^{(\rho)}\boldsymbol{\Sigma}_{[1\cdots r]}^{(\rho)}(\mathbf{V}^{(\rho)})^T = \mathbf{R}$ be the SVD of $\mathbf{R}$. Similarly we use $\hat{\mathbf{R}} := \mathbf{C}^{-1/2}\hat{\mathbf{P}}\tilde{\mathbf{C}}^{-1/2}$ with SVD $\hat{\mathbf{U}}^{(\rho)}\hat{\boldsymbol{\Sigma}}^{(\rho)}(\hat{\mathbf{V}}^{(\rho)})^T = \hat{\mathbf{R}}$. Let $\tilde{\mathbf{R}} := \hat{\mathbf{U}}^{(\rho)}\hat{\boldsymbol{\Sigma}}_{[1\cdots r]}^{(\rho)}(\hat{\mathbf{V}}^{(\rho)})^T$ be the best rank $r$ approximation of $\hat{\mathbf{R}}$. Use $\mathbb{O}_{r \times r}$ to denote set of all $r \times r$ orthogonal matrices, let $\mathbf{O} \in \mathbb{O}_{r \times r}$, using triangle inequality we have:

$$\|\boldsymbol{\Psi}(x) - \boldsymbol{\Psi}(z)\| = \|\mathbf{O}\boldsymbol{\Psi}(x) - \mathbf{O}\boldsymbol{\Psi}(z)\|$$
$$\le \|\mathbf{O}\boldsymbol{\Psi}(x) - \hat{\boldsymbol{\Psi}}(x)\| + \|\hat{\boldsymbol{\Psi}}(x) - \hat{\boldsymbol{\Psi}}(z)\| + \|\mathbf{O}\boldsymbol{\Psi}(z) - \hat{\boldsymbol{\Psi}}(z)\|, \tag{86}$$

this yields:

$$dist(x,z) - \widehat{dist}(x,z) = \|\boldsymbol{\Psi}(x) - \boldsymbol{\Psi}(z)\| - \|\hat{\boldsymbol{\Psi}}(x) - \hat{\boldsymbol{\Psi}}(z)\| \tag{87}$$

$$\le \|\mathbf{O}\boldsymbol{\Psi}(x) - \hat{\boldsymbol{\Psi}}(x)\| + \|\mathbf{O}\boldsymbol{\Psi}(z) - \hat{\boldsymbol{\Psi}}(z)\|. \tag{88}$$

Similarly we can get:

$$\widehat{dist}(x,z) - dist(x,z) = \|\hat{\boldsymbol{\Psi}}(x) - \hat{\boldsymbol{\Psi}}(z)\| - \|\boldsymbol{\Psi}(x) - \boldsymbol{\Psi}(z)\| \tag{89}$$

$$\leq \|\mathbf{O}\boldsymbol{\Psi}(x) - \hat{\boldsymbol{\Psi}}(x)\| + \|\mathbf{O}\boldsymbol{\Psi}(z) - \hat{\boldsymbol{\Psi}}(z)\|. \tag{90}$$

Therefore, taking infimum over $\mathbf{O} \in \mathbb{O}_{r \times r}$ we have:

$$\left| dist(x,z) - \widehat{dist}(x,z) \right| \leq \inf_{\mathbf{O} \in \mathbb{O}_{r \times r}} \|\mathbf{O}\boldsymbol{\Psi}(x) - \hat{\boldsymbol{\Psi}}(x)\| + \|\mathbf{O}\boldsymbol{\Psi}(z) - \hat{\boldsymbol{\Psi}}(z)\| \tag{91}$$

$$= \inf_{\mathbf{O} \in \mathbb{O}_{r \times r}} \|\Phi(x)^T \mathbf{C}^{-1/2}(\mathbf{U}^{(\rho)}\boldsymbol{\Sigma}^{(\rho)}_{[1\cdots r]}\mathbf{O}^T - \hat{\mathbf{U}}^{(\rho)}\hat{\boldsymbol{\Sigma}}^{(\rho)}_{[1\cdots r]})\|$$

$$+ \|\Phi(z)^T \mathbf{C}^{-1/2}(\mathbf{U}^{(\rho)}\boldsymbol{\Sigma}^{(\rho)}_{[1\cdots r]}\mathbf{O}^T - \hat{\mathbf{U}}^{(\rho)}\hat{\boldsymbol{\Sigma}}^{(\rho)}_{[1\cdots r]})\| \tag{92}$$

$$\leq \inf_{\mathbf{O} \in \mathbb{O}_{r \times r}} 2L_{max}^{1/2}\|\mathbf{U}^{(\rho)}\boldsymbol{\Sigma}^{(\rho)}_{[1\cdots r]}\mathbf{O}^T - \hat{\mathbf{U}}^{(\rho)}\hat{\boldsymbol{\Sigma}}^{(\rho)}_{[1\cdots r]}\| \tag{93}$$

$$= \inf_{\mathbf{O} \in \mathbb{O}_{r \times r}} 2L_{max}^{1/2}\|\mathbf{R}\mathbf{V}^{(\rho)}\mathbf{O}^T - \tilde{\mathbf{R}}\hat{\mathbf{V}}^{(\rho)}\| \tag{94}$$

$$= \inf_{\mathbf{O} \in \mathbb{O}_{r \times r}} 2L_{max}^{1/2}\|\mathbf{R}(\mathbf{V}^{(\rho)}\mathbf{O}^T - \hat{\mathbf{V}}^{(\rho)}) + (\mathbf{R} - \tilde{\mathbf{R}})\hat{\mathbf{V}}^{(\rho)}\| \tag{95}$$

$$\leq \inf_{\mathbf{O} \in \mathbb{O}_{r \times r}} 2L_{max}^{1/2}\left(\|\mathbf{R}\| \cdot \|\mathbf{V}^{(\rho)}\mathbf{O}^T - \hat{\mathbf{V}}^{(\rho)}\| + \|\mathbf{R} - \tilde{\mathbf{R}}\| \cdot \|\hat{\mathbf{V}}^{(\rho)}\|\right) \tag{96}$$

$$= \inf_{\mathbf{O} \in \mathbb{O}_{r \times r}} 2L_{max}^{1/2}\left(\|\mathbf{R}\| \cdot \|\mathbf{V}^{(\rho)}\mathbf{O}^T - \hat{\mathbf{V}}^{(\rho)}\| + \|\mathbf{R} - \tilde{\mathbf{R}}\|\right) \tag{97}$$

$$\leq 2L_{max}^{1/2}(\|\mathbf{R}\|\sqrt{2}\|sin\Theta(\mathbf{V}^{(\rho)}, \hat{\mathbf{V}}^{(\rho)})\| + \|\mathbf{R} - \tilde{\mathbf{R}}\|) \tag{98}$$

$$\leq 2L_{max}^{1/2}(\sqrt{2}\|\mathbf{R}\|\frac{\|\mathbf{R} - \tilde{\mathbf{R}}\|}{\sigma_r(\mathbf{R})} + \|\mathbf{R} - \tilde{\mathbf{R}}\|) \tag{99}$$

$$\leq 2L_{max}^{1/2}(1 + \sqrt{2}\kappa(\mathbf{R}))\|\mathbf{R} - \tilde{\mathbf{R}}\| \tag{100}$$

$$\leq 4L_{max}^{1/2}(1 + \sqrt{2}\kappa(\mathbf{R}))\|\mathbf{R} - \hat{\mathbf{R}}\| \tag{101}$$

$$= 4L_{max}^{1/2}(1 + \sqrt{2}\kappa(\mathbf{R}))\|\mathbf{C}^{-1/2}(\mathbf{P} - \hat{\mathbf{P}})\tilde{\mathbf{C}}^{-1/2}\| \tag{102}$$

$$\leq 4\sqrt{\frac{L_{max}}{\rho_N \tilde{\rho}_N}}\left[1 + \sqrt{2}\kappa(\mathbf{R})\right]\|\mathbf{P} - \hat{\mathbf{P}}\|, \tag{103}$$

where from (100) to (101) we use Weyl's inequality [Wey12] and the fact that rank of $\tilde{\mathbf{R}}$ is $r$ to get $\sigma_{r+1}(\hat{\mathbf{R}}) \leq \|\mathbf{R} - \hat{\mathbf{R}}\|$, therefore,

$$\|\mathbf{R} - \tilde{\mathbf{R}}\| \leq \|\mathbf{R} - \hat{\mathbf{R}}\| + \|\tilde{\mathbf{R}} - \hat{\mathbf{R}}\| = \|\mathbf{R} - \hat{\mathbf{R}}\| + \sigma_{r+1}(\hat{\mathbf{R}}) \leq 2\|\mathbf{R} - \hat{\mathbf{R}}\|. \tag{104}$$

Note that $\mathbf{R}$ is the coefficient matrix of $p(y|x)$ in expansion with bases $\{\frac{\Phi_i(\cdot)}{\sqrt{\rho_i}}\}_{i=1}^N \times \{\frac{\tilde{\Phi}_i(\cdot)}{\sqrt{\tilde{\rho}_i}}\}_{i=1}^N$ using $L^2(\pi) \times L^2$ inner product, equivalently it is coefficient matrix of $\sqrt{\pi(x)}p(y|x)$ in expansion with bases $\{\frac{\Phi_i(\cdot)\sqrt{\pi(\cdot)}}{\sqrt{\rho_i}}\}_{i=1}^N \times \{\frac{\tilde{\Phi}_i(\cdot)}{\sqrt{\tilde{\rho}_i}}\}_{i=1}^N$ using $L^2 \times L^2$ inner product. By assumption 2, $\sqrt{\pi(x)}p(y|x)$ can be represented using $\{\frac{\Phi_i(\cdot)\sqrt{\pi(\cdot)}}{\sqrt{\rho_i}}\}_{i=1}^N \times \{\frac{\tilde{\Phi}_i(\cdot)}{\sqrt{\tilde{\rho}_i}}\}_{i=1}^N$, therefore we have $\kappa\left(\sqrt{\pi(x)}p(y|x)\right) = \kappa(\mathbf{R})$. We conclude proof using Eqn.(11). $\qquad\square$

## B.3  Proof of Theorem 3

*Proof.* We show that $\|p(\cdot|\cdot) - \hat{p}(\cdot|\cdot)\|_{L^2(\pi) \times L^2} = \|\mathbf{R} - \tilde{\mathbf{R}}\|_F$. From Lemma 6 and definition of $\hat{p}(y|x)$ we have:

$$p(y|x) = \Phi(x)^T \mathbf{C}^{-1}\mathbf{P}\tilde{\mathbf{C}}^{-1}\tilde{\Phi}(y) = \Phi(x)^T \mathbf{C}^{-1/2}\mathbf{R}\tilde{\mathbf{C}}^{-1/2}\tilde{\Phi}(y), \tag{105}$$

$$\hat{p}(y|x) = \Phi(x)^T \mathbf{C}^{-1/2}\tilde{\mathbf{R}}\tilde{\mathbf{C}}^{-1/2}\tilde{\Phi}(y). \tag{106}$$

Then we have:

$$p(y|x) - \hat{p}(y|x) = \Phi(x)^T \mathbf{C}^{-1/2} (\mathbf{R} - \tilde{\mathbf{R}}) \tilde{\mathbf{C}}^{-1/2} \tilde{\Phi}(y). \tag{107}$$

Recall that $\Upsilon(\cdot) := \mathbf{C}^{-1/2} \Phi(\cdot)$ is a vector of orthonormal functions in $L^2(\pi)$ and $\tilde{\Upsilon}(\cdot) := \tilde{\mathbf{C}}^{-1/2} \tilde{\Phi}(\cdot)$ is a vector of orthonormal functions in $L^2$. Then we have:

$$\|p(\cdot|\cdot) - \hat{p}(\cdot|\cdot)\|_{L^2(\pi) \times L^2} = \|\Phi(\cdot)^T \mathbf{C}^{-1/2} (\mathbf{R} - \tilde{\mathbf{R}}) \tilde{\mathbf{C}}^{-1/2} \tilde{\Phi}(\cdot)\| \tag{108}$$

$$= \|\Upsilon(\cdot)(\mathbf{R} - \tilde{\mathbf{R}}) \tilde{\Upsilon}(\cdot)\|_{L^2(\pi) \times L^2} \tag{109}$$

$$= \|\mathbf{R} - \tilde{\mathbf{R}}\|_F. \tag{110}$$

Combining (104) with (110) yields

$$\|p(\cdot|\cdot) - \hat{p}(\cdot|\cdot)\|_{L^2(\pi) \times L^2} = \|\mathbf{R} - \tilde{\mathbf{R}}\|_F \tag{111}$$

$$\leq \sqrt{r} \|\mathbf{R} - \tilde{\mathbf{R}}\| \tag{112}$$

$$\leq 2\sqrt{r} \|\mathbf{R} - \hat{\mathbf{R}}\| \tag{113}$$

$$= 2\sqrt{r} \|\mathbf{C}^{-1/2} (\mathbf{P} - \hat{\mathbf{P}}) \tilde{\mathbf{C}}^{-1/2}\| \tag{114}$$

$$\leq 2\sqrt{\frac{r}{\rho_N \tilde{\rho}_N}} \|\mathbf{P} - \hat{\mathbf{P}}\|. \tag{115}$$

We conclude the proof upon using (11). $\qquad \square$

## C    Proof of Results in Section 5

### C.1    Technical Lemmas

**Lemma 7.** *Under Assumption 1-2, for each $q_i^*(\cdot)$, it can be written as:*

$$q_i^*(\cdot) = \sum_{k=1}^r z_{ik} v_k(\cdot), \tag{116}$$

*where $v_k(\cdot)$ are the right singular functions for $p(\cdot|\cdot)$. Each $q_i^*(\cdot)$ is a probability density function.*

*Proof.* $\{\Omega_i^*\}_{i=1}^m$ forms the best partition in terms of solving k-means problem. Then on each $\Omega_i^*$, we must have $q_i^*(\cdot)$ solves the problem

$$\min_{q_i(\cdot) \in \tilde{\mathcal{H}}} \int_{\Omega_i^*} \pi(x) \|p(\cdot|x) - q_i(\cdot)\|_{L^2}^2 dx. \tag{117}$$

This is solved by

$$q_i^*(\cdot) = \frac{1}{\pi(\Omega_i^*)} \int_{\Omega_i^*} \pi(x) p(\cdot|x) dx. \tag{118}$$

To show $q_i^*(\cdot)$ is probability distribution, note that $q_i^*(y) \geq 0$ for all $y$ because $p(y|x) \geq 0$ for all $y$ and $x$. Furthermore, we have:

$$\int_{\Omega^*} q_i^*(y) dy = \frac{1}{\pi(\Omega_i)} \int_\Omega \int_{\Omega_i^*} \pi(x) p(y|x) dx dy \tag{119}$$

$$= \frac{1}{\pi(\Omega_i^*)} \int_{\Omega_i^*} \int_\Omega \pi(x) p(y|x) dy dx \tag{120}$$

$$= \frac{1}{\pi(\Omega_i^*)} \int_{\Omega_i^*} \pi(x) dx \tag{121}$$

$$= 1. \tag{122}$$

Without loss of generality, we assume that decomposition $p(y|x) = \sum_{i=1}^{r} \sigma_k u_k(x) v_k(x)$ in Assumption 1 is the SVD, *i.e.*,

$$\sigma_k = \mathbf{\Sigma}_{k,k}^{(\rho)}, \quad u_k(\cdot) = \left(\mathbf{U}_{[:,k]}^{(\rho)}\right)^T \mathbf{C}^{-1/2}\Phi(\cdot), \quad v_k(\cdot) = \left(\mathbf{V}_{[:,k]}^{(\rho)}\right)^T \tilde{\mathbf{C}}^{-1/2}\tilde{\Phi}(\cdot).$$

To prove Eqn.(116), we plug in SVD of $p(y|x)$ into Eqn.(118):

$$q_i^*(\cdot) = \frac{1}{\pi(\Omega_i^*)} \int_{\Omega_i^*} \pi(x) p(\cdot|x) dx \tag{123}$$

$$= \sum_{k=1}^{r} \frac{\sigma_k}{\pi(\Omega_i^*)} \int_{\Omega_i^*} \pi(x) u_k(x) v_k(\cdot) dx. \tag{124}$$

Taking $z_{ik} := \frac{\sigma_k}{\pi(\Omega_i^*)} \int_{\Omega_i^*} \pi(x) u_k(x) dx$, we finish the proof. $\qquad\square$

Next lemma is key to prove Theorem 5. Before proving the lemma, we define a function $T(\cdot|\cdot)$ that represents the perturbation of $p(\cdot|x)$ from its closest probability distribution $q_i^*(\cdot)$:

$$T(\cdot|x) = \sum_{i=1}^{m} \mathbb{1}_{\Omega_i^*}(x)\left(p(\cdot|x) - q_i^*(\cdot)\right). \tag{125}$$

By definition of $\Delta_2^2$ we know that:

$$\|T(\cdot|\cdot)\|_{L^2(\pi)\times L^2}^2 = \Delta_2^2. \tag{126}$$

Equivalently, one can rewrite the k-means problem in $\mathbb{R}^r$ using the $\mathbf{\Psi}(\cdot)$ coordinate as:

$$\min_{(\Omega_1,\cdots,\Omega_m)} \min_{s_1,\cdots,s_k\in\mathbb{R}^r} \sum_{i=1}^{m} \int_{\Omega_i} \pi(x)\|\mathbf{\Psi}(x) - s_i\|_{l_2}^2 dx.$$

We showed in Lemma 6 that $s_i^* = [z_{i1}, \cdots, z_{ir}]^T$, then we construct function $E(\cdot) : \Omega \to \mathbb{R}^r$ by:

$$E(x) = \sum_{i=1}^{m} \mathbb{1}_{\Omega_i^*}(x)(\mathbf{\Psi}(x) - [z_{i1}, \cdots, z_{ir}]^T). \tag{127}$$

It is easy to verify that

$$\mathbf{\Psi}(x) = \mathbf{Z}\theta(x) + E(x), \tag{128}$$

where $\mathbf{Z} = [z_{ij}]_{r\times m}$ is given in Lemma 5, $\theta := [\mathbb{1}_{\Omega_1^*}, \cdots, \mathbb{1}_{\Omega_m^*}]^T$, moreover, since $E(\cdot)$ is the counterpart of $T(\cdot|x)$ in $\mathbb{R}^r$, we have:

$$\|E(\cdot)\|_{L^2(\pi)}^2 = \Delta_2^2. \tag{129}$$

We further define following quantities which are useful for the statement of the lemma:

$$\delta_k^2 := \min_{l\neq k} \|q_l^* - q_k^*\|_{L^2}^2 = \min_{1\neq k} \|\mathbf{Z}_{*l} - \mathbf{Z}_{*k}\|, \tag{130}$$

$$\overline{\mathbf{\Psi}}(x) := \hat{\mathbf{Z}}\hat{\theta}(x) \qquad \hat{\mathbf{Z}}_{*i} = \hat{s}_i, \qquad \hat{\theta}(x) := [\mathbb{1}_{\hat{\Omega}_1^*}, \cdots, \mathbb{1}_{\hat{\Omega}_m^*}]^T. \tag{131}$$

We use $\mathbb{O}_{r\times r}$ to denote the set of all $r \times r$ orthogonal matrices. For any $\mathbf{O} \in \mathbb{O}_{r\times r}$, we define:

$$S_k(\mathbf{O}\overline{\mathbf{\Psi}}) := \{x \in \Omega_k^* : \|\mathbf{O}\overline{\mathbf{\Psi}}(x) - \mathbf{Z}_{*k}\| \geq \delta_k/2\}. \tag{132}$$

For the ease of notation, we will use $S_k$ instead of $S_k(\mathbf{O}\overline{\mathbf{\Psi}})$. For a vector valued function $A(\cdot) : \Omega \to \mathbb{R}^r$, let its $L^2(\pi)$ norm to be $\|A(\cdot)\|_{L^2(\pi)} := \left(\int_\Omega \pi(x)\|A(x)\|_{\ell_2}^2 dx\right)^{1/2}$.

**Lemma 8.** *With quantities defined above, for any $\mathbf{O} \in \mathbb{O}_{r\times r}$ we have:*

$$\sum_{k=1}^{m} \pi(S_k)\delta_k^2 \leq 16\left(\|\mathbf{O}\hat{\mathbf{\Psi}}(\cdot) - \mathbf{\Psi}(\cdot)\|_{L^2(\pi)} + \|E(\cdot)\|_{L^2(\pi)}\right)^2 \tag{133}$$

$$= 16\left(\|\mathbf{U}^{(\rho)}\mathbf{\Sigma}_{[1\cdots r]}^{(\rho)}\mathbf{O} - \hat{\mathbf{U}}^{(\rho)}\hat{\mathbf{\Sigma}}_{[1\cdots r]}^{(\rho)}\|_F + \Delta_2\right)^2. \tag{134}$$

*In addition, if for any $1 \leq k \leq m$ we have:*

$$\frac{16\Big(\|\boldsymbol{U}^{(\rho)}\boldsymbol{\Sigma}_{[1\cdots r]}^{(\rho)}\boldsymbol{O} - \hat{\boldsymbol{U}}^{(\rho)}\hat{\boldsymbol{\Sigma}}_{[1\cdots r]}^{(\rho)}\|_F + \Delta_2\Big)^2}{\delta_k^2} < \pi(\Omega_k^*). \tag{135}$$

*Then every data point on $G := \cup_{k=1}^{r}(\Omega_k^* \setminus S_k)$ is correctly classified.*

*Proof.* We follow the proof of Lemma 5.3 in [LR$^+$15]. By definition of $S_k$, we have:

$$\sum_{k=1}^{m} \pi(S_k)\delta_k^2 \leq 4\sum_{k=1}^{m} \int_{S_k} \pi(x)\|\mathbf{O}\overline{\boldsymbol{\Psi}}(x) - \mathbf{Z}_{*k}\|^2 dx \tag{136}$$

$$\leq 4\int_{\Omega} \pi(x)\|\mathbf{O}\overline{\boldsymbol{\Psi}}(x) - \mathbf{Z}\theta(x)\|^2 dx \tag{137}$$

$$= 4\big\|\mathbf{O}\overline{\boldsymbol{\Psi}}(\cdot) - \mathbf{Z}\theta(\cdot)\big\|_{L^2(\pi)}^2. \tag{138}$$

To bound $\big\|\mathbf{O}\overline{\boldsymbol{\Psi}}(\cdot) - \mathbf{Z}\theta(\cdot)\big\|_{L^2(\pi)}$:

$$\big\|\mathbf{O}\overline{\boldsymbol{\Psi}}(\cdot) - \mathbf{Z}\theta(\cdot)\big\|_{L^2(\pi)}$$

$$\leq \big\|\mathbf{O}\overline{\boldsymbol{\Psi}}(\cdot) - \mathbf{O}\hat{\boldsymbol{\Psi}}(\cdot)\big\|_{L^2(\pi)} + \big\|\mathbf{O}\hat{\boldsymbol{\Psi}}(\cdot) - \boldsymbol{\Psi}(\cdot)\big\|_{L^2(\pi)} + \big\|\boldsymbol{\Psi}(\cdot) - \mathbf{Z}\theta(\cdot)\big\|_{L^2(\pi)} \tag{139}$$

$$= \big\|\overline{\boldsymbol{\Psi}}(\cdot) - \hat{\boldsymbol{\Psi}}(\cdot)\big\|_{L^2(\pi)} + \big\|\mathbf{O}\hat{\boldsymbol{\Psi}}(\cdot) - \boldsymbol{\Psi}(\cdot)\big\|_{L^2(\pi)} + \big\|E(\cdot)\big\|_{L^2(\pi)}, \tag{140}$$

where we use the fact that $\boldsymbol{\Psi}(x) = \mathbf{Z}\theta(x) + E(x)$. Because $\overline{\boldsymbol{\Psi}}(x) = \hat{\mathbf{Z}}\hat{\theta}(\cdot)$ solves the empirical k-means problem,

$$\big\|\overline{\boldsymbol{\Psi}}(\cdot) - \hat{\boldsymbol{\Psi}}(\cdot)\big\|_{L^2(\pi)} \leq \big\|\mathbf{O}^T\mathbf{Z}\theta(\cdot) - \hat{\boldsymbol{\Psi}}(\cdot)\big\|_{L^2(\pi)} = \big\|\mathbf{Z}\theta(\cdot) - \mathbf{O}\hat{\boldsymbol{\Psi}}(\cdot)\big\|_{L^2(\pi)}$$

$$\leq \big\|\mathbf{Z}\theta(x) - \boldsymbol{\Psi}(\cdot)\big\|_{L^2(\pi)} + \big\|\mathbf{O}\hat{\boldsymbol{\Psi}}(\cdot) - \boldsymbol{\Psi}(\cdot)\big\|_{L^2(\pi)} \tag{141}$$

$$= \big\|E(\cdot)\big\|_{L^2(\pi)} + \big\|\mathbf{O}\hat{\boldsymbol{\Psi}}(\cdot) - \boldsymbol{\Psi}(\cdot)\big\|_{L^2(\pi)}.$$

Plugging (141) into (140) gives

$$\big\|\mathbf{O}\overline{\boldsymbol{\Psi}}(\cdot) - \mathbf{Z}\theta(\cdot)\big\|_{L^2(\pi)} \leq 2\|\mathbf{O}\hat{\boldsymbol{\Psi}}(x) - \boldsymbol{\Psi}(x)\|_{L^2(\pi)} + 2\|E(\cdot)\|_{L^2(\pi)}. \tag{142}$$

It follows from Eqn.(138) that

$$\sum_{k=1}^{m} \pi(S_k)\delta_k^2 \leq 16\Big(\|\mathbf{O}\hat{\boldsymbol{\Psi}}(x) - \boldsymbol{\Psi}(x)\|_{L^2(\pi)} + \|E(\cdot)\|_{L^2(\pi)}\Big)^2 \tag{143}$$

$$= 16\Big(\|\mathbf{O}\hat{\boldsymbol{\Psi}}(x) - \boldsymbol{\Psi}(x)\|_{L^2(\pi)} + \Delta_2\Big)^2. \tag{144}$$

We then show: $\|\mathbf{O}\hat{\boldsymbol{\Psi}}(x) - \boldsymbol{\Psi}(x)\|_{L^2(\pi)}^2 = \|\mathbf{U}^{(\rho)}\boldsymbol{\Sigma}_{[1\cdots r]}^{(\rho)}\mathbf{O} - \hat{\mathbf{U}}^{(\rho)}\hat{\boldsymbol{\Sigma}}_{[1\cdots r]}^{(\rho)}\|_F^2$:

$$\|\mathbf{O}\hat{\boldsymbol{\Psi}}(x) - \boldsymbol{\Psi}(x)\|_{L^2(\pi)}^2 = \|\hat{\boldsymbol{\Psi}}(x) - \mathbf{O}^T\boldsymbol{\Psi}(x)\|_{L^2(\pi)}^2 \tag{145}$$

$$= \int_{\Omega} \pi(x)\Phi(x)^T\mathbf{C}^{-1/2}(\mathbf{U}^{(\rho)}\boldsymbol{\Sigma}_{[1\cdots r]}^{(\rho)}\mathbf{O} - \hat{\mathbf{U}}^{(\rho)}\hat{\boldsymbol{\Sigma}}_{[1\cdots r]}^{(\rho)})$$

$$\cdot (\mathbf{U}^{(\rho)}\boldsymbol{\Sigma}_{[1\cdots r]}^{(\rho)}\mathbf{O} - \hat{\mathbf{U}}^{(\rho)}\hat{\boldsymbol{\Sigma}}_{[1\cdots r]}^{(\rho)})^T\mathbf{C}^{-1/2}\Phi(x)dy \tag{146}$$

$$= Tr\Big((\mathbf{U}^{(\rho)}\boldsymbol{\Sigma}_{[1\cdots r]}^{(\rho)}\mathbf{O} - \hat{\mathbf{U}}^{(\rho)}\hat{\boldsymbol{\Sigma}}_{[1\cdots r]}^{(\rho)})(\mathbf{U}^{(\rho)}\boldsymbol{\Sigma}_{[1\cdots r]}^{(\rho)}\mathbf{O} - \hat{\mathbf{U}}^{(\rho)}\hat{\boldsymbol{\Sigma}}_{[1\cdots r]}^{(\rho)})^T\Big) \tag{147}$$

$$= \|\mathbf{U}^{(\rho)}\boldsymbol{\Sigma}_{[1\cdots r]}^{(\rho)}\mathbf{O} - \hat{\mathbf{U}}^{(\rho)}\hat{\boldsymbol{\Sigma}}_{[1\cdots r]}^{(\rho)}\|_F^2. \tag{148}$$

Plug this back into Eqn.(143)

$$\sum_{k=1}^{m} \pi(S_k)\delta_k^2 \leq 16\Big(\|\mathbf{U}^{(\rho)}\boldsymbol{\Sigma}_{[1\cdots r]}^{(\rho)}\mathbf{O} - \hat{\mathbf{U}}^{(\rho)}\hat{\boldsymbol{\Sigma}}_{[1\cdots r]}^{(\rho)}\|_F + \Delta_2\Big)^2. \tag{149}$$

which finishes the proof of Eqn.(133).

We then prove if condition (135) holds, then every data point on $G := \cup_{k=1}^{r}(\Omega_k^* \setminus S_k)$ is correctly classified. From Eqn.(133):

$$\pi(S_k)\delta_k^2 \leq \sum_{k=1}^{m} \pi(S_k)\delta_k^2 \leq 16\Big(\|\mathbf{U}^{(\rho)}\mathbf{\Sigma}_{[1\cdots r]}^{(\rho)}\mathbf{O} - \hat{\mathbf{U}}^{(\rho)}\hat{\mathbf{\Sigma}}_{[1\cdots r]}^{(\rho)}\|_F + \Delta_2\Big)^2. \tag{150}$$

If condition (135) holds, dividing $\delta_k^2$ on both sides of Eqn.(150) gives:

$$\pi(S_k) \leq \frac{16\Big(\|\mathbf{O}\hat{\mathbf{\Psi}}(x) - \mathbf{\Psi}(x)\|_{L^2(\pi)} + \Delta_2\Big)^2}{\delta_k^2} < \pi(\Omega_k^*). \tag{151}$$

From this we know $T_k := \Omega_k^* \setminus S_k \neq \emptyset$ for all $k$. We then prove data on $T_k$ are correctly classified for any $k$. If $x \in T_k, y \in T_l$ for $k \neq l$, we must have $\overline{\mathbf{\Psi}}(x) \neq \overline{\mathbf{\Psi}}(y)$, otherwise we have

$$\max(\delta_k, \delta_l) \leq \|\mathbf{Z}_{*k} - \mathbf{Z}_{*l}\|_{l_2} \leq \|\mathbf{Z}_{*k} - \mathbf{O}\overline{\mathbf{\Psi}}(x)\| + \|\mathbf{Z}_{*l} - \mathbf{O}\overline{\mathbf{\Psi}}(y)\| < \delta_k/2 + \delta_l/2$$

which is impossible. On the other hand, of $x, y \in T_k$ for some $k$, then we must have $\overline{\mathbf{\Psi}}(x) = \overline{\mathbf{\Psi}}(y)$, otherwise $\overline{\mathbf{\Psi}}(x)$ will take more than $m$ values which is impossible due to is definition in Eqn.(131). $\qquad\square$

## C .2    Proof of Theorem 4

*Proof.* For ease of notation we omit the perturbation $\sigma$. Let $\mathbb{O}_{r \times r}$ be the set of all $r \times r$ orthogonal matrices. Let $\mathbf{O} \in \mathbb{O}_{r \times r}$ be an $r \times r$ orthogonal matrix that will be specified later. Denote $\mathbf{R} = \mathbf{U}^{(\rho)}\mathbf{\Sigma}_{[1\cdots r]}^{(\rho)}(\mathbf{V}^{(\rho)})^T = \mathbf{C}^{-1/2}\mathbf{P}\tilde{\mathbf{C}}^{-1/2}$ and $\tilde{\mathbf{R}} = \hat{\mathbf{U}}^{(\rho)}\hat{\mathbf{\Sigma}}_{[1\cdots r]}^{(\rho)}(\hat{\mathbf{V}}^{(\rho)})^T$. From Eqn.(134) we have:

$$\sum_{k=1}^{m} \pi(S_k)\delta_k^2 \leq 16\Big(\|\mathbf{U}^{(\rho)}\mathbf{\Sigma}_{[1\cdots r]}^{(\rho)}\mathbf{O} - \hat{\mathbf{U}}^{(\rho)}\hat{\mathbf{\Sigma}}_{[1\cdots r]}^{(\rho)}\|_F + \Delta_2\Big)^2 \tag{152}$$

$$= 16\Big(\|\mathbf{R}\mathbf{V}^{(\rho)}\mathbf{O} - \tilde{\mathbf{R}}\hat{\mathbf{V}}^{(\rho)}\|_F + \Delta_2\Big)^2. \tag{153}$$

To bound $\|\mathbf{R}\mathbf{V}^{(\rho)}\mathbf{O} - \tilde{\mathbf{R}}\hat{\mathbf{V}}^{(\rho)}\|_F$:

$$\|\mathbf{R}\mathbf{V}^{(\rho)}\mathbf{O} - \tilde{\mathbf{R}}\hat{\mathbf{V}}^{(\rho)}\|_F = \|\mathbf{R}(\mathbf{V}^{(\rho)}\mathbf{O} - \hat{\mathbf{V}}^{(\rho)}) + (\mathbf{R} - \tilde{\mathbf{R}})\hat{\mathbf{V}}^{(\rho)}\|_F \tag{154}$$

$$\leq \|\mathbf{R}\| \cdot \|\mathbf{V}^{(\rho)}\mathbf{O} - \hat{\mathbf{V}}^{(\rho)}\|_F + \|\mathbf{R} - \tilde{\mathbf{R}}\| \cdot \|\hat{\mathbf{V}}^{(\rho)}\|_F \tag{155}$$

$$= \|\mathbf{R}\| \cdot \|\mathbf{V}^{(\rho)}\mathbf{O} - \hat{\mathbf{V}}^{(\rho)}\|_F + \sqrt{r}\|\mathbf{R} - \tilde{\mathbf{R}}\|, \tag{156}$$

where on the last equality we use the fact that $\hat{\mathbf{V}}^{(\rho)}$ is orthogonal hence $\|\hat{\mathbf{V}}^{(\rho)}\|_F = \sqrt{r}$. We know that there exists some $\mathbf{O} \in \mathbb{O}_{r \times r}$ such that

$$\|\mathbf{V}^{(\rho)}\mathbf{O} - \hat{\mathbf{V}}^{(\rho)}\|_F \leq \sqrt{2}\|sin\mathbf{\Theta}(\mathbf{V}^{(\rho)}, \hat{\mathbf{V}}^{(\rho)})\|_F \leq \sqrt{2r}\|sin\mathbf{\Theta}(\mathbf{V}^{(\rho)}, \hat{\mathbf{V}}^{(\rho)})\|. \tag{157}$$

By Wedin's lemma and the fact that $\mathbf{R}$ and $\tilde{\mathbf{R}}$ have rank $r$ we know that

$$\|\mathbf{V}^{(\rho)}\mathbf{O} - \hat{\mathbf{V}}^{(\rho)}\|_F \leq \frac{\sqrt{2r}\|\mathbf{R} - \tilde{\mathbf{R}}\|}{\sigma_r(\mathbf{R})}. \tag{158}$$

Plug this back into Eqn.(156) we have:

$$\|\mathbf{R}\mathbf{V}^{(\rho)}\mathbf{O} - \tilde{\mathbf{R}}\hat{\mathbf{V}}^{(\rho)}\|_F \leq (\sqrt{r} + \frac{\sqrt{2r}\|\mathbf{R}\|}{\sigma_r(\mathbf{R})})\|\mathbf{R} - \tilde{\mathbf{R}}\| \tag{159}$$

$$\leq 2\sqrt{2r}\kappa(\mathbf{R})\|\mathbf{R} - \tilde{\mathbf{R}}\| \tag{160}$$

$$\leq 4\sqrt{2r}\kappa(\mathbf{R})\|\mathbf{R} - \hat{\mathbf{R}}\| \tag{161}$$

$$\leq 4\sqrt{\frac{2r}{\rho_N \tilde{\rho}_N}}\kappa(\mathbf{R})\|\mathbf{P} - \hat{\mathbf{P}}\| \tag{162}$$

$$= 4\sqrt{\frac{2r}{\rho_N \tilde{\rho}_N}}\kappa\|\mathbf{P} - \hat{\mathbf{P}}\|, \tag{163}$$

where in the last equality we use the fact that $\kappa(\mathbf{R}) = \kappa\left(\sqrt{\pi(x)}p(y|x)\right)$ which we proved in the proof of Theorem 2.

According to Lemma 4, there exists a constant $c > 0$ such that if

$$n \geq c \cdot \frac{\kappa^2 r \bar{\lambda} t_{mix} \log(2 t_{mix} N / \delta)}{\rho_N \tilde{\rho}_N} \cdot \max\left\{ \frac{1}{(\Delta_1/4 - \Delta_2)^2}, \frac{2}{\epsilon \Delta_1^2}, \frac{4\Delta_2^2}{\epsilon^2 \Delta_1^4} \right\}, \tag{164}$$

then with probability at least $1 - \delta$,

$$\|\hat{\mathbf{P}} - \mathbf{P}\| \leq \frac{1}{4\kappa(\mathbf{R})} \sqrt{\frac{\rho_N \tilde{\rho}_N}{2r}} \min\left\{ \Delta_1/4 - \Delta_2, \sqrt{\epsilon/2}\Delta_1, \frac{\epsilon \Delta_1^2}{2\Delta_2} \right\}. \tag{165}$$

Under condition (165), $16\left(\|\mathbf{RV}^{(\rho)}\mathbf{O} - \hat{\mathbf{R}}\hat{\mathbf{V}}^{(\rho)}\|_F + \Delta_2\right)^2 < \Delta_1^2$, which ensures (135) is true hence we can use Lemma 8. We also have $\|\mathbf{RV}^{(\rho)}\mathbf{O} - \hat{\mathbf{R}}\hat{\mathbf{V}}^{(\rho)}\|_F^2 \leq \epsilon\Delta_1^2/2$ and $\|\mathbf{RV}^{(\rho)}\mathbf{O} - \hat{\mathbf{R}}\hat{\mathbf{V}}^{(\rho)}\|_F \Delta_2 \leq \epsilon\Delta_1^2/2$. These two inequalities together imply

$$16\left(\|\mathbf{RV}^{(\rho)}\mathbf{O} - \hat{\mathbf{R}}\hat{\mathbf{V}}^{(\rho)}\|_F + \Delta_2\right)^2 \tag{166}$$

$$= 16\left(\|\mathbf{RV}^{(\rho)}\mathbf{O} - \hat{\mathbf{R}}\hat{\mathbf{V}}^{(\rho)}\|_F^2 + 2\|\mathbf{RV}^{(\rho)}\mathbf{O} - \hat{\mathbf{R}}\hat{\mathbf{V}}^{(\rho)}\|_F \Delta_2 + \Delta_2^2\right) \tag{167}$$

$$\leq \epsilon\Delta_1^2 + 16\Delta_2^2. \tag{168}$$

We can now derive an upper bound for the misclassification rate $M$:

$$M(\hat{\Omega}_1^*, \cdots, \hat{\Omega}_m^*) \leq \sum_{k=1}^{m} \frac{\pi(S_k)}{\pi(\Omega_k^*)} \tag{169}$$

$$\leq \sum_{k=1}^{m} \frac{\pi(S_k)\delta_k^2}{\Delta_1^2} \tag{170}$$

$$\leq \frac{16\left(\|\mathbf{RV}^{(\rho)}\mathbf{O} - \hat{\mathbf{R}}\hat{\mathbf{V}}^{(\rho)}\|_F + \Delta_2\right)^2}{\Delta_1^2} \tag{171}$$

$$\leq \epsilon + \frac{16\Delta_2^2}{\Delta_1^2}. \tag{172}$$

$\square$

# D   Experiment with DQN

The game of Demon Attack is simulated in the Arcade Learning Environment ([BNVB13]), which provides an interface to hundreds of Atari 2600 games and serves an important testbed for deep reinforcement learning algorithms. We closely follow the experimental setting, network structure and training method used by [MKS+15]. In this environment, each game frame $o_t$ is a $210 \times 160 \times 3$ image. In each interactive step the agent takes in the last 16 frames and preprocesses them to be the input state $s_t = \phi(\{o_{t-i}\}_{i=0}^{15})$. The state $s_t$ is an $84 \times 84 \times 4$ rescaled, grey-scale image, and is the input to the neural network $Q(s_t, \cdot; \theta)$. The first convolution layer in the network has 32 filters of size 8 stride 4, the second layer has 64 layers of size 4 stride 2, the final convolution layer has 64 filters of size 3 stride 1, and is followed by a fully-connected hidden layer of 512 units. The output is another fully-connected layer with six units that correspond to the six action values $\{Q(s_t, a_i; \theta)\}_{i=1}^{6}$. The agent selects an action based on these state-action values, repeats the selected action four times, observes four subsequent frames $\{o_{t+i}\}_{i=1}^{4}$ and receives an accumulated reward $r_t$.

The agent in the DQN algorithm "learns" through a novel variant of Q-learning that employs the techniques of target net ($\theta^-$) and experience replay ($D$) ([MKS+15]), and conducts gradient descent to the following loss function at each iteration:

$$\mathcal{L}(\theta) = \mathbf{E}_{s,a,r,s' \sim U(D)}\left[\left(r + \gamma \max_{a'} Q(s', a'; \theta^-) - Q(s, a; \theta)\right)^2\right].$$

While our training regime and hyper-parameters are almost the same as those of [MKS+15], we use Adam optimizer ([KB15]) with a decaying learning rate, and a smaller replay buffer of size 500k frames. Training is done over 2.5 million steps, i.e., 10 million game frames. The Q-network is stored and evaluated every 500k steps. The best policy among these evaluations attains a 150-200% human-level performance (which was reported in [MKS+15]), and is later used as our sampling policy.

The raw input to the state embedding algorithm is a time series of length 47936 and dimension 512, comprising 130 trajectories generated by the fully-connected hidden layer in DQN when it is running the sampling policy. The embeddings are obtained through the same process as in Experiment 6.1 with a Gaussian kernel and 200 random Fourier features. The rank $r$ is set to be 3, and the time interval $\tau$ corresponds to 12 game frames, i.e., 0.36 second in real time. Both the raw date and the embeddings are projected onto 2D planes by t-SNE with a perplexity of 40.