[Reviews · NeurIPS 2019]

Reviewer 1



This appears to be a detailed, carefully written, and theoretically/empirically supported piece of work. The results are based on RKHS theory and appear to extend the state-of-the-art in a number of ways, including learning KMEs while dealing with dependent samples (e.g. from a Markov chain), low dimensional/rank approximations with state-of-the-art error bounds, and a variational approach for clustering, as well as a robust set of experiments. On the whole, I am satisfied that this clearly meets the acceptance criteria for NeurIPS and I am recommending it for acceptance. I have a few comments below on aspects of the paper, where things were either unclear, or might benefit from a revisit. It is unclear how strong assumption 2 is. It seems to me that if the marginal p(x) is in the Hilbert space H, and the joint p(x,y) is in the joint Hilbert space Hx\tilde{H} then it is natural to assume that the conditional is too, but it isn't clear whether this is a strong or weak assumption. When could it be violated? The equations under [lines 147 and 158] seem to be inconsistent (one is the theoretical and the other the approximate conditional probability). I think the first equation should have half powers on the C-matrices. Theorem 4 appears to assume that the underlying process has a correct clustering of states, but this isn't explicit in the theorem. The stochastic diffusion process could be a bit better described. And a better interpretation/intuition over the results. The colours in the #clusters=15 plot should be distinct (I count 8 colours and 17 contiguous regions). There appears to be a bit of repetition in the description of the DQN experiment, and the pairs of states could be more formally described (how did you identify that these states should be close to one another before you conducted the experiment, or was it a post-hoc analysis). ## typos [line 48] metastablet sets -> metastable sets [line 130-1] shift-invariance kernel -> shift-invariant kernel [line 158] Assumptions 2 -> Assumption 2 [line 168] states within the same set shares -> states within the same set share Equation under [line 179] The symbol i on the top line should be an x I think. Also, it should be stated that this misclassification rate is with respect to data where the ground truth is that the states belong to m clusters. [line 220] with the number of clusters vary -> with a varying number of clusters ## After reading the reviews I can see some valid points made by the other reviewers in their comments particularly with respect to the clarity of what contributions are novel. I still consider this to be a good paper though, and I will keep my score as it stands.

Reviewer 2



Aside from some minor grammatical issues, I found this to be an interesting paper introducing new methods that leveraging low-rank structure to embed transition distributions and time-varying states. I think the idea of the diffusion distance can inspire further work in distance metrics for sequential models as well as new learning paradigms. (i) Where do the authors see their work fitting in relative to existing work on Koopman operators and Dynamic Mode Decomposition. For example in https://papers.nips.cc/paper/6713-learning-koopman-invariant-subspaces-for-dynamic-mode-decomposition.pdf, the authors use the level sets of the Koopman eigenfunction to find basins of attraction of the dynamics (Figure 5 -- right in the pdf) in a manner similar to what is done here via clustering. (ii) Have you experimented with clustering the raw time-series (i.e. without the state embedding). What do those results look like and how do they qualitatively differ from what is presented in Figure 2? (iii) Could you comment on the assumption in line 171 that the clustering admits a unique optimal solution. What properties of the time-series do you think might be necessary for that assumption to be satisfied? (iv) How did the quality of the state embeddings (as visualized in Figure 2) vary as a function of the number of Fourier features used. Presumably there is some tradeoff between the number of features needed to capture the properties of the dynamics to a sufficient degree and the complexity of the underlying dynamics (for example, the number of basins of attraction and how close they are). Have you experimented with this?

Reviewer 3



I think the theory proposed in section 2 is quite interesting and a novel way of computing a Markov transition matrix given sampled transitions. From my understanding it basically requires sampling transitions and projecting them on to a random set of features, and then using the projection to approximate a transition matrix and its SVD. I think that the paper is not written very well because it does not emphasize the intuition behind projecting to random features or other work on random features. Further this could be modified to be a streaming algorithm which the authors don't do. The next section is more confusing. First, the authors don't specify in what sense their markov transition matrix is irreversible. I believe they are referring to the fact that the normal diffusion operator is reversible after multiplication by a degree conjugate. However, it seems like the same general type of symmetrization trick is working here. At least this needs to be written clear for a non-familiar audience.

Reviewer 4



After the rebuttal, I have increased my score accordingly. ################################ The authors propose a kernel mean embedding nonparametric estimation for low-rank Markov chains, and provide corresponding statistical rate. Majors: 1. Nonparametric estimation for Markov chain probability kernel seems to have been considered in [1]. And [1] assume a more general model with an approximately low-rank structure with exponential decay of the eigenvalue. Minimax estimate rate is also derived. Can the authors clarify the novelty beyond [1]? 2. I feel assumption 2 is strong since the authors assume the kernel function for Hilbert space can be exactly represented by a finite sum of basis functions. Does this mean that there is no approximation error any more? Typically in RHKS literature, people represent the kernel function by an infinite sum spectral expansion. Different kernel functions then are categorized by different eigenvalue decay rates, e.g. polynomial decay kernel or exponential decay kernel. 3. The rate in Theorem 1 is just a parametric rate that roughly the same as the one in [2][3]. So I am wondering how hard to generalize the technique from [2][3] to this paper. 4. I appreciate the authors could implement their method on Atari game. But I am not convinced that why you take the last hidden layer of DQN? What is the benefit and will it improve the score of the game? The explanation for some particular actions of a game is quire artificial since we know vanilla DQN itself is not stable. The author should test some datasets to show that if the transition kernel is indeed low-rank. Minor: 1. The comparison with plain KME is not fair since reshaped KME uses additional information, saying the knowledge of low-rankness. [1]. Matthias Löffler, Antoine Picard. Spectral thresholding for the estimation of Markov chain transition operators. [2]. Anru Zhang, Mengdi Wang. Spectral State Compression of Markov Processes. [3]. Xudong Li, Mengdi Wang, Anru Zhang. Estimation of Markov Chain via Rank-Constrained Likelihood.

[Author Response · NeurIPS 2019]

**Response to Reviewer #1**

● *"I would like to see Theorem 4 reworded. It assumes that the underlying process has a correct clustering of states?"*

**RE:** You are right. Thm 4 assumes there is an underlying partition that attains the smallest value of distortion (eq(4)). The minimal distortion can be nonzero and the optimal solution may not be exactly correct. We will reword Thm 4 to make it easier to interpret.

● *"How to find state pairs in DQN analysis"* **RE:** We computed and ranked pairwise embedding distances for 2000 randomly picked states. Then we screened the top 100 closest pairs and pick those with large raw-data distances.

● *"... carefully written, theoretically/empirically supported ... The paper is already of a very high standard in my opinion ... clarity, readability.."* **RE:** Thanks! We are excited to have your support, and we will make an effort to improve readability of the paper.

**Response to Reviewer #2**

● *Relation with existing work on Koopman and dynamic model decomposition (DMD)*:

**RE:** Thank you for mentioning the related paper (Takeishi etal 18). We will cite it and add more discussion about DMD. DMD and our method share the same spirit - finding low-dimensional subspace of the transition function/kernel using decomposition. For comparison, DMD is originally developed for linear systems and then generalized to nonlinear dynamics. Our focus is different - we focus on the probabilistic transition of unstructured stochastic process and statistical error from finite dependent data. Our method applies to randomly jumping process (eg discrete-state games). We hope our analysis will inspire new developments on DMD.

● *"In Figure 2, experiment with clustering raw time series (without embedding) and compare."*
**RE:** Clustering raw time series produces partitions that look like grids and loses the temporal information (see the new Figure (*) for comparison). Figure 3 in the submission also gives a useful comparison and shows that embedding improves the clusters.

(*)15 clusters of states. Left: raw data;
Right: after embeddings (Fig 2 in paper).

● *"Uniqueness of optimal solution required by Thm 4. What properties of the time-series are needed"*

**RE:** The uniqueness is a theoretical simplification. In practice, it means that the process admits a low-dim block structure or internal physical state, which is often unique.

● *Dependency on number of random features:* **RE:** According to (Rahimi&Recht 08), we pick the number of features to be $O(\frac{d}{\epsilon^2})$ to approximate the RKHS ($\epsilon$ is a user-picked accuracy level). Experiments suggest more than $O(\frac{d}{\epsilon^2})$ features doesn't add value.

**Response to Reviewer #3**

● *"Not written very well because it does not emphasize the intuition behind projecting to random features"*

**RE:** The results are based on projection to general RKHS space, not limited to random features. One needs random features only if the RKHS is known but explicit bases functions are unknown/infinite. In this case, randomizing features to approximate RKHS is a standard technique with rich theory developed by (Rahimi and Recht 2008).

● *"Explanation of the reversibility of normal diffusion maps and the idea here ... Not everyone knows diffusion map."*

**RE:** Thanks for the comment. We should have explained better, but were limited by the page limit.

– Irreversibility means we do *not* assume reversibility of the Markov process (a restricted technical condition), so the results apply to most practical time series. It also means that the frequency matrix/operator is asymmetric and doesn't admit eigendecomposition. Therefore typical analysis no longer applies, and proving the theorems requires nontrivial work.

– Diffusion map is a standard technique for dimension reduction (see Coifman etal 05, 06, Nedlar etal 09). Although it is related, it is not a prerequisite for understanding our paper. Our main idea is spectral decomposition of the transition kernel projected onto an RKHS space. We will make an effort to improve the writing and explanation.

● *"Potential for streaming the proposed algorithm"*

**RE:** Good point. It is possible to make the algorithm streaming (eg. using the Oja's method), but is beyond the current scope. (Our Algorithm 1 makes a single pass over time series. It runs in $O(d^2)$ space and $O(nd^2 + dr)$ total time, which is quite efficient.)

**Response to Reviewer #4**

● *Comparison to [1],[2],[3] (which were cited in our submission).*

**RE:** [1] assumes reversibility of the underlying Markov chain and wavalet bases; its analysis relies on eigendocomposition and eigenvalue decay condition. In contrast, our method applies to practical time series (typically not reversible) and general RKHS, where eigendecomposition of $P$ doesn't exist. [2,3] work for finite-state time series and lack scalability. Our method leverages kernel information and applies to multi-variate (or even unstructured) time series. Thm 1 generalizes [2,3]. Thms 2,3,4 provide theory for nonparametric estimation of metastable sets and preserving diffusion distance, which were not available in the literature.

● *"Assumption 2 is strong ... exactly represented by a finite sum of basis functions ... no approximation error ..."*

**RE:** The conditions are for simplicity and the analysis can be generalized to include approximation error. Computers operate in low dimensions. So we leverage (RR08) to approximate the kernel space and compute nonparametric estimation in finite dimensions.

● *"Why take the last layer of DQN? What is the benefit and will it improve the score? ... DQN is not stable. "*

**RE:** Very good questions. We should have better explained: Our DQN is pre-trained and only used as a static feature map for reducing the dimension of raw images. Our experiment can be viewed as using the composition between neural tangent kernel (Jacot etal 18) and Gaussian kernel. This experiment is to visualize and interpret state embeddings from game trajectories. Improving the score is beyond our scope and is an interesting direction for future work.

(a) 10 clusters by [2] (b) 10 embedding clusters (c) Singular value decay

● **"Additional experiments. Justify the low-rankness."** We further conduct a taxi-trip experiment (same setting as in [2]), using Gaussian kernel for pickup/dropoff locations. Figure (a) is the result from [2] without using kernel information; (b) is our clusters using kernelized state embeddings; (c) illustrates the singular value decay of transition matrix and justifies the low-rank assumption. In (a), the clusters are often mixed up and overlapping. In (b), state embedding leads to more meaningful zones and higher granularity, using the same data size. Due to the rebuttal's space limit, we can't explain every detail. We hope this additional experiment is convincing and we will add more details in the final paper.

[Meta-Review · NeurIPS 2019]

After reading the author's rebuttal, all four reviewers provide accepting scores. Reviewer 4 increased his/her score despite not being fully convinced by the justification of the low rank assumption on the transition probability matrix. Overall there is agreement among the reviewers that the paper makes an interesting contribution and they value that the authors could run their method in a complex environment like Atari. The AC encourages the authors to include the suggestions made by the reviewers (in particular those from reviewer 3 and 4) for the camera ready version of the paper.